# Noisy Adaptation Generates Lévy Flights in Attractor Neural Networks

**Xingsi Dong**[1,2,3,*], **Tianhao Chu**[1,2,3,*], **Tiejun Huang**[4], **Zilong Ji**[1,2,3,†], **Si Wu**[1,2,3,†]

1, School of Psychology and Cognitive Sciences, Peking Univerisity.
2, IDG/McGovern Institute for Brain Research, Peking University.
3, PKU-Tsinghua Center for Life Sciences,
Academy for Advanced Interdisciplinary Studies, Peking University.
4, School of Electronics Engineering and Computer Science, Peking University.
*: Equal contributions. †: Corresponding authors.
{dxs19980605,chutianhao,tjhuang,jizilong,siwu}@pku.edu.cn

## Abstract

Lévy flights describe a special class of random walks whose step sizes satisfy a power-law tailed distribution. As being an efficient searching strategy in unknown environments, Lévy flights are widely observed in animal foraging behaviors. Recent studies further showed that human cognitive functions also exhibit the characteristics of Lévy flights. Despite being a general phenomenon, the neural mechanism at the circuit level for generating Lévy flights remains largely unclear. Here, we investigate how Lévy flights can be achieved in attractor neural networks. To elucidate the underlying mechanism clearly, we first study continuous attractor neural networks (CANNs), and find that noisy neural adaptation, exemplified by spike frequency adaptation (SFA) in this work, can generate Lévy flights representing transitions of the network state in the attractor space. Specifically, the strength of SFA defines a travelling wave boundary, below which the network state displays local Brownian motion, and above which the network state displays long-jump motion. Noises in neural adaptation cause the network state to intermittently switch between these two motion modes, manifesting the characteristics of Lévy flights. We further extend the study to a general attractor neural network, and demonstrate that our model can explain the Lévy-flight phenomenon observed during free memory retrieval of humans. We hope that this study will give us insight into understanding the neural mechanism for optimal information processing in the brain.

## 1 Introduction

Lévy flights, also termed anomalous diffusion or super diffusion, refer to a special class of random walks whose step sizes follow a distribution with a long power-law tail. Mathematically, this power-law tailed distribution is expressed as

$$p(x) \sim x^{-1-\alpha}, \tag{1}$$

where the step size $x$ can be measured either in the spatial domain, e.g., the length of a movement, or in the temporal domain, e.g., the time interval between successive events. The parameter $\alpha$ is called the Lévy exponent satisfying $0 < \alpha < 2$. When $\alpha \geq 2$, the above process degenerates to Brownian motion due to the Central Limit Theorem [1]. Compared to Brownian motion, whose step sizes satisfy a Gaussian distribution, Lévy flights are much more likely to generate large step sizes. Thus, it is often intuitively stated that Lévy flights are composed of frequent local motion (similar to the Brownian motion) and intermittent long-jump motion (Fig. 1). In contrast to Brownian motion

which tends to oversample a local area, Lévy flights with long jumps provide a more efficient way to search for scarce targets that are randomly distributed in an unknown environment, and the highest search efficiency occurs at $\alpha = 1$ [2, 3, 4].

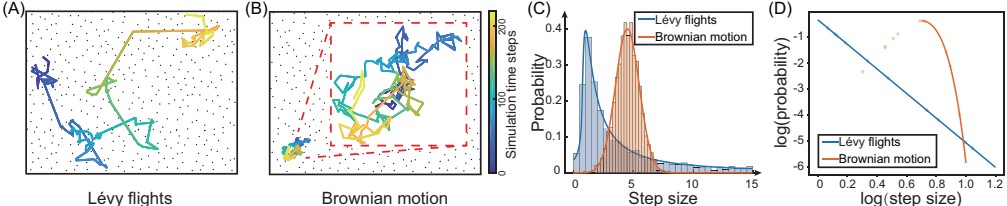

Figure 1: Illustrations of Lévy flights and Brownian motion. (A) An example of Lévy flights generated by sampling step sizes from Eq. (1), with $\alpha$ chosen between 0 and 2. (B) An example of Brownian motion generated by sampling step sizes from Eq. (1), with a $\alpha$ larger than 2. Both Lévy flights and Brownian motion showed here are generated with the same simulation duration, i.e., 200 time steps. (C) The histograms of step sizes ($x$ in Eq. (1)) of Lévy flights (blue) and Brownian motion (orange), respectively. Notably, Lévy flights have ultra-long jumps which are absent in Brownian motion. (D) The log-log plot of the histograms of step sizes in (C) shows that the distribution of Lévy flights follows a power-law tail.

As being an efficient strategy for information search, Lévy flights have been widely observed in foraging behaviors across different animal species, ranging from microzooplankton [5] to drosophila [6], and from albatrosses [7] to spider monkeys [8]. Similar foraging-like patterns are also observed in human behaviors, e.g., individual mobility in the geographical scale [9]. Experimental studies further revealed that Lévy flights also exist in human cognitive functions. For instance, it has been found that the long tailed distribution holds for the saccadic eye movement in a random quenched salience field [10], for the time interval between successive retrieved items in a free memory retrieval task [11], and for the mental exploration of 'bid space' in Lowest Unique Bid Auctions (LUBA) [12]. Recently, an fMRI study unveiled that the dynamics of neural activation at the resting state of the brain can also be characterized by Lévy flights [13]. In a more detailed study, Pfeiffer and Foster showed that the awake reply trajectories of the hippocampal place cell ensemble in immobile rats resemble superdiffusion dynamics [14], which is very similar to Lévy flights. Intriguingly, Lévy flights have also attracted considerable attention in the deep learning community for modeling the optimization process with the stochastic gradient descent (SGD) [15, 16, 17].

Despite that Lévy flights have been widely observed and that various computational models have been proposed to account for their occurrences in different scenarios (see, e.g., [18]), the neural basis for the brain generating Lévy flights remains far from resolved (see more discussions in Sec. 4). In the present study, we will investigate the neural mechanism for generating Lévy flights at the circuit level, i.e., how a neural network with appropriate biological features generates Lévy flights in the space of information presentation. We hope that this study will help us to further understand the emergence of Lévy flights in cognitive functions of humans.

Specifically, we will study attractor neural networks which model how information is represented in neural systems [19, 20], and hence Lévy flights in the corresponding attractor spaces reflect information processing in the brain. To elucidate the underlying mechanism clearly, we focus on studying continuous attractor neural networks (CANNs), as this allows us to solve the network dynamics analytically. The key characteristic of Lévy flights is the occurrence of intermittent long jumps. In order to achieve this property, we consider noisy adaptation in the neural dynamics, which is exemplified by spike frequency adaptation (SFA) in this work. The role of adaptation is to induce a spontaneously moving state, called travelling wave, in the network dynamics, which occurs when the strength of SFA exceeds a threshold. Thus, if the mean SFA strength is set slightly below the threshold, noises in adaptation will occasionally push the SFA strength to exceed the threshold. Consequently, the network state will temporarily fall into the travelling wave state and experience a long-jump movement in the attractor space. Over time, noisy adaptation causes the network state to switch intermittently between local motion (when the SFA strength remains below the threshold) and long-jump motion in the attractor space, manifesting the characteristic of Lévy flights. We carry out theoretical analyses to formally verify the above mechanism, and simulation results agree with

our theoretical analyses very well. We further extend the study to a general Hopfield-like attractor network, and use it to model the Lévy-flight phenomenon as observed in the free memory retrieval experiment [11]. We hope that this study will help us to understand how neural circuits generate Lévy flights, and give us insight into understanding the emergence and the computational roles of Lévy flights in brain functions.

## 2   Lévy flights in continuous attractor neural networks

We first study the generation of Lévy flights in continuous attractor neural networks (CANNs). CANNs have been widely used as canonical models for elucidating the encoding of continuous features in neural systems [21, 22, 23, 24], including, for instance, the orientation [25], the head direction [26], and the spatial location [27, 28]. The advantage of studying CANNs is that it allows us to solve the network dynamics analytically [29] and hence gives us insight into understanding the computational mechanism of generating Lévy flights. In Sec. 3, we will extend the theoretical findings to other attractor networks.

### 2.1   The model

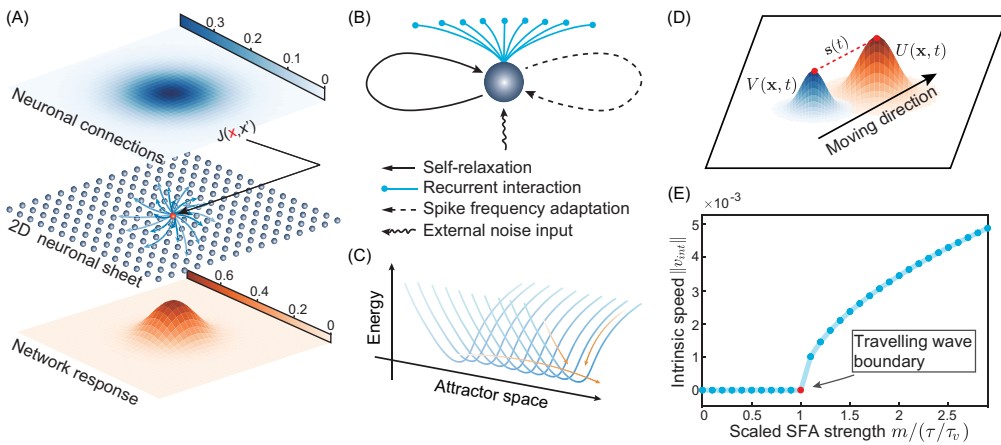

Figure 2: A two-dimensional (2D) continuous attractor neural network (CANN) with neural adaptation. (A) Illustrating the network structure. Top: the excitatory connections between the neuron at $\mathbf{x}$ and neurons at other locations. Middle: the two-dimensional neuronal sheet on which neurons are uniformly distributed. Bottom: the network response with a Gaussian bell shape. (B) The kinetics of a single neuron, determined by the self-relaxation, recurrent interaction, spike frequency adaptation, and external input. (C) the attractor space formed by all stationary states of the network, on which the CANN is neutrally stable (for clearance, the case of 1D is shown). (D) The network state with SFA as a negative feedback in the 2-D CANN feature space. The bump $U(\mathbf{x}, t)$ can move spontaneously in the attractor space in any direction depending on the initial state. The bump of the adaptation current $V(\mathbf{x}, t)$ lags behind $U(\mathbf{x}, t)$ with a separation $\mathbf{s}(t)$ along the moving direction, due to the delayed feedback modulation of SFA. (E) The intrinsic speed of the travelling wave $||\mathbf{v}_{int}||$ vs. the SFA strength (scaled by $\tau/\tau_v$). The red point represents the travelling wave boundary with $m = \tau/\tau_v$.

As illustrated in Fig. 2A, we consider a two-dimensional CANN, in which neurons are uniformly distributed on a rectangular neuronal sheet according to their preferred feature values. Denote $U(\mathbf{x}, t)$ as the synaptic input to the neuron at $\mathbf{x}$, with $\mathbf{x} = (x_1, x_2)$ and $x_1, x_2 \in (-\infty, \infty)$, and $r(\mathbf{x}, t)$ the corresponding firing rate. The dynamics of $U(\mathbf{x}, t)$ is determined by its own relaxation, the recurrent inputs from other neurons, the neural adaptation, and the external noise input (Fig. 2B), which is written as,

$$\tau \frac{\partial U(\mathbf{x}, t)}{\partial t} = -U(\mathbf{x}, t) + \rho \int_{\mathbf{x}'} J(\mathbf{x}, \mathbf{x}') r(\mathbf{x}', t) \, d\mathbf{x}' - V(\mathbf{x}, t) + \sigma_U \xi_U(\mathbf{x}, t). \tag{2}$$

Here, $\tau$ is the synaptic time constant and $\rho$ the neuronal density. The term $\sigma_U \xi_U(\mathbf{x}, t)$ represents the external noise input, with $\sigma_U$ the noise strength and $\xi_U(\mathbf{x}, t)$ the Gaussian white noise of zero

mean and unit variance. In this study, we propose an internal mechanism to generate Lévy flights, therefore we only consider noisy inputs of zero mean to the network and do not consider non-zero drift inputs (see Sec. 4 for more discussions). The recurrent neuronal connections, $J(\mathbf{x}, \mathbf{x}') = J_0/(2\pi a^2) \exp\left[-\|\mathbf{x} - \mathbf{x}'\|^2/(2a^2)\right]$, with $\|\mathbf{x} - \mathbf{x}'\|^2 = (x_1 - x_1')^2 + (x_2 - x_2')^2$, are translation-invariant on the neuronal sheet (Fig. 2A), which means that $J(\mathbf{x}, \mathbf{x}')$ is a function of $\|\mathbf{x} - \mathbf{x}'\|$. The nonlinear relationship between the firing rate $r(\mathbf{x}, t)$ and the synaptic input $U(\mathbf{x}, t)$ is implemented by divisive normalization, which is written as,

$$r(\mathbf{x}, t) = \frac{U^2(\mathbf{x}, t)}{1 + k\rho \int_{\mathbf{x}'} U^2(\mathbf{x}', t)\, d\mathbf{x}'}. \tag{3}$$

Here, the parameter $k$ controls the normalization strength. In reality, divisive normalization could be implemented by shunting inhibition [30].

The term $V(\mathbf{x}, t)$ on the right-hand side of Eq. (2) represents an adaptive current. Adaptation is a general phenomenon referring to that a neuron population generates negative feedback to suppress its response when the activity level is high. Neural adaptation can result from different mechanisms, and their effects on the network dynamics are similar (see Sec. 4 for more discussions). Here, we consider spike frequency adaptation (SFA) at the single neuron level as an example. In reality, SFA could be implemented by the interplay between calcium currents and intracellular calcium dynamics with calcium-gated potassium channels [31]. The dynamics of $V(\mathbf{x}, t)$ is written as,

$$\tau_v \frac{\partial V(\mathbf{x}, t)}{\partial t} = -V(\mathbf{x}, t) + [m + \sigma_m \xi_m(\mathbf{x}, t)] U(\mathbf{x}, t), \tag{4}$$

where $\tau_v$ is the time constant of SFA, with $\tau_v \gg \tau$, indicating that SFA is a process much slower than neuronal firing. $m$ is the mean SFA strength, with $\sigma_m$ the noise strength and $\xi_m(\mathbf{x}, t)$ denoting the Gaussian white noise of zero mean and unit variance.

We first review the properties of the CANN without adaptation, i.e., by setting $m = 0$ and $\sigma_m = 0$ in Eq. (4). It has been shown that without the external noise input ($\sigma_U = 0$), the CANN holds a continuous family of Gaussian-shaped stationary states, called bumps, when the global inhibition amplitude $k$ is smaller than a critical value $k_c = \rho J_0^2/(32\pi a^2)$ [29]. These bump states are expressed as $\overline{U}(\mathbf{x}) = A_U \exp\left[-\|\mathbf{x} - \mathbf{z}\|^2/(4a^2)\right]$, where $\mathbf{z}$ is a free parameter representing the bump position, i.e., a feature value encoded by the network, and $A_U$ is a constant representing the bump height. These bump states form an attractor space (Fig. 2C), on which the CANN is neutrally stable, which means that a small external input can drive the bump to move smoothly in the attractor space without distorting its shape. This neutral stability is the key that enables CANNs to realize accurate path integration [27, 28], smooth tracking of a moving object [32], and efficient population decoding [33]. Specifically, under the drive of external Gaussian noises, the bump will exhibit Brownian motion in the attractor space [34].

We further review the properties of the CANN with a fixed adaptation strength (i.e., $m$ is a constant and $\sigma_m = 0$). It has been shown that without external input, the CANN holds a moving bump as its stationary state when $m > \tau/\tau_v$, which is called travelling wave (Fig. 2E) [35]. This state reflects the intrinsic mobility of the network, as the bump moves spontaneously in the attractor space without relying on an external drive. The travelling wave state is expressed as $\overline{U}(\mathbf{x}, t) = A_U' \exp\left[-(\mathbf{x} - \mathbf{v}_{int}t)^2/(4a^2)\right]$, with $\|\mathbf{v}_{int}\| = 2a/\tau_v\sqrt{m\tau_v/\tau - \sqrt{m\tau_v/\tau}}$ the travelling speed and $A_U'$ the bump height [35]. The mechanism underlying this intrinsic mobility is intuitively understandable. Suppose that the bump initially appears at an arbitrary location in the attractor space. Due to SFA, those most active neurons, i.e., those around the peak of the bump, receive the strongest adaptation current, and their activities are suppressed the most. Since those neighboring neurons are less active and hence are less suppressed, the bump tends to move away to the neighborhood due to recurrent connections and competitions via divisive normalization; after moving to the new location, the suppression and competition start again. As a result, the bump will keep moving in the attractor space. Analogous to the situation without adaptation, if the SFA strength is fixed and $m < \tau/\tau_v$, external noises will only induce Brownian motion of the bump in the attractor space.

## 2.2 Noisy adaptation generates Lévy flights

Before going to the detailed mathematical analyses, we first provide an intuitive understanding of how noisy adaptation leads to Lévy flights. As introduced above, a CANN with SFA has the intrinsic

mobility of generating a travelling wave when the SFA strength is sufficiently strong. The condition of $m = \tau/\tau_v$ defines the travelling wave boundary. Above the boundary, the bump will move spontaneously in the attractor space; below the boundary, the bump will either remain static if no external input exists, or exhibit Brownian motion when external noises are applied. The interesting phenomenon emerges if the SFA strength is noisy and its mean value $m$ is close to the boundary. In such a case, fluctuations (due to the noise term $\sigma_m \xi(\mathbf{x}, t)$ in Eq. (4)) will push the SFA strength to cross the boundary occasionally, which causes the network to fall into the travelling wave state temporarily, and consequently, the bump travels over a long distance (a long jump) in the attractor space. Over time, along with fluctuations of the SFA strength, the bump displays intermittent local motion (when the SFA strength remains below the threshold) and long-jump motion, manifesting the characteristic of Lévy flights. We present the formal analyses below.

Consider that the noise strengths in Eqs. (2&4) are sufficiently small, such that their effects on distorting the bump shape are negligible, and we assume that the network state has the Gaussian form as in the static case, which is given by,

$$U(\mathbf{x}, t) = A_u(t) \exp\left\{ -\frac{[\mathbf{x} - \mathbf{z}(t)]^2}{4a^2} \right\}, \tag{5}$$

$$r(\mathbf{x}, t) = A_r(t) \exp\left\{ -\frac{[\mathbf{x} - \mathbf{z}(t)]^2}{2a^2} \right\}, \tag{6}$$

$$V(\mathbf{x}, t) = A_v(t) \exp\left\{ -\frac{[\mathbf{x} - (\mathbf{z}(t) - \mathbf{s}(t))]^2}{4a^2} \right\}, \tag{7}$$

where $A_u(t)$, $A_r(t)$ and $A_v(t)$ represent the heights of bumps $U(\mathbf{x}, t)$, $r(\mathbf{x}, t)$ and $V(\mathbf{x}, t)$, respectively. $\mathbf{z}(t)$ is the center of bumps $U(\mathbf{x}, t)$ and $r(\mathbf{x}, t)$, whose trajectory reflects changes of the network state in the attractor space. $\mathbf{z}(t) - \mathbf{s}(t)$ is the center of bump $V(\mathbf{x}, t)$, with $\mathbf{s}(t)$ denoting the separation between $U(\mathbf{x}, t)$ and $V(\mathbf{x}, t)$. Note that the bump $V(\mathbf{x}, t)$ always lags behind $U(\mathbf{x}, t)$, reflecting that the adaptation current is delayed with respect to the neural response (Fig. 2D).

Previous works have shown that the dynamics of a CANN is dominated by very few motion modes [29, 36]. Therefore, to solve the network dynamics, we can project the network dynamics onto those dominating modes and simplify the analyses significantly[1]. In the current study, we adopt the first two dominating motion modes, corresponding to the changes of bump height and position, respectively, which are given by,

$$u_0(\mathbf{x}|\mathbf{z}) = \frac{1}{a\sqrt{2\pi}} \exp\left\{ -\frac{[\mathbf{x} - \mathbf{z}(t)]^2}{4a^2} \right\}, \tag{8}$$

$$u_1(\mathbf{x}|\mathbf{z}) = \frac{1}{a^2\sqrt{2\pi}} [\mathbf{x} - \mathbf{z}(t)] \exp\left\{ -\frac{[\mathbf{x} - \mathbf{z}(t)]^2}{4a^2} \right\}. \tag{9}$$

By projecting the network dynamics onto these two modes, we obtain the dynamics of the bump height and position, and the latter reflects how the network state changes in the attractor space. Specifically, by substituting the presumptive network state Eqs. (5-7) into the network dynamics Eqs. (2&4), and then projecting them on the motion mode of bump height Eq. (8), we obtain the dynamics of bump heights (see SI.1 for the details), which are written as,

$$\tau \frac{dA_u}{dt} = -A_u - A_v + \frac{J_0 \rho A_r}{2} + \frac{\sigma_U}{a\sqrt{2\pi}} \xi_{U,0}(t), \tag{10}$$

$$\tau_v \frac{dA_v}{dt} = -A_v + mA_u + \frac{\sigma_m A_u}{2a\sqrt{\pi}} \xi_{m,0}(t), \tag{11}$$

where $\xi_{U,0}(t)$ and $\xi_{m,0}(t)$ denote, respectively, the projected noises of $\xi_U(t)$ and $\xi_m(t)$ on the bump height mode, which are still Gaussian white noises of zero mean and unit variance. Note that we do not explicitly write down the dynamics of $A_r$, as it has a deterministic relationship with $A_u$, that is, by substituting Eqs. (5-6) into Eq. (3), we obtain $A_r = A_u^2/(1 + 2\pi a^2 k\rho A_u^2)$.

Similarly, substituting the network state Eqs. (5-7) into the network dynamics Eqs. (2&4), and then projecting them on the motion mode for bump position Eq. (9), we obtain the dynamics of bump

---

[1]Projecting a function $f(\mathbf{x})$ on a mode $u(\mathbf{x})$ means computing $\int_{\mathbf{x}} f(\mathbf{x})u(\mathbf{x})d\mathbf{x}$.

positions (see SI.1 for the details), which are,

$$\tau \frac{d\mathbf{z}}{dt} = \frac{A_v}{A_u}\mathbf{s} + \frac{\sigma_U}{A_u}\sqrt{\frac{2}{\pi}}\xi_{U,1}(t), \tag{12}$$

$$\tau_v \frac{d\mathbf{s}}{dt} = \left(\frac{\tau_v A_v}{\tau A_u} - \frac{m A_u}{A_v} - \frac{\sigma_m A_u \xi_{m,0}}{2 A_v \sqrt{\pi a}}\right)\mathbf{s} + \frac{\tau_v \sigma_U}{\tau A_u}\sqrt{\frac{2}{\pi}}\xi_{U,1}(t) - \frac{\sigma_m A_u}{A_v}\sqrt{\frac{1}{2\pi}}\xi_{m,1}. \tag{13}$$

Here $\xi_{U,1}(t)$ and $\xi_{m,1}(t)$ denote, respectively, the projected noises of $\xi_U(t)$ and $\xi_m(t)$ on the bump position mode, which are also Gaussian white noises of zero mean and unit variance.

We can obtain the stationary distributions of $A_u$ and $A_v$ in Eqs. (10&11) by solving the corresponding Fokker-Planck equations, which is a general method to describe the time evolution of the probabilistic density function of a moving particle in physics. Moreover, since $\sigma_U$ and $\sigma_m$ are very small, the variances of $A_u$ and $A_v$ can be ignored compared to their mean values. Under this approximation, Eqs. (12-13) can be further simplified by replacing $A_u$ and $A_v$ with their mean values $\tilde{A}_u$ and $\tilde{A}_v$ and using the approximation $\tilde{A}_v = m\tilde{A}_u$ according to Eq. (11), which are written as,

$$\tau \frac{d\mathbf{z}}{dt} = m\mathbf{s} + \frac{\sigma_U}{\tilde{A}_u}\sqrt{\frac{2}{\pi}}\xi_{U,1}(t), \tag{14}$$

$$\tau_v \frac{d\mathbf{s}}{dt} = -\left[1 - \frac{\tau_v}{\tau}m + \frac{\sigma_m}{2\sqrt{\pi}am}\xi_{m,0}(t)\right]\mathbf{s} + \sqrt{\frac{2}{\pi}\left(\frac{\tau_v \sigma_U}{\tau \tilde{A}_u}\right)^2 + \frac{1}{2\pi}\left(\frac{\sigma_m}{m}\right)^2}\xi_{\mathbf{s}}(t). \tag{15}$$

Here $\xi_s(t)$ is a newly defined Gaussian white noise of zero mean and unit variance (by combining the last two noise terms in Eq. (13)). Eq. (14) shows that the bump position $\mathbf{z}(t)$ is determined by a drift term reflecting the contribution of SFA and a diffusion term reflecting the contribution of the external noise input. Apparently, when no adaptation exists, i.e., no drift in Eq. (14) ($m = 0$), the bump movement is Brownian motion due to the diffusion noise.

To solve the dynamics of the bump position $\mathbf{z}(t)$, it is necessary to first solve the dynamics of $\mathbf{s}(t)$ expressed in Eq. (15). We see that the drift coefficient of $\mathbf{s}(t)$ in Eq. (15) consists of two parts. The first part, $1 - m\tau_v/\tau$, measures the distance of the mean SFA strength to the travelling wave boundary (given by $\tau/\tau_v$), and we denote $\mu = 1 - m\tau_v/\tau$ as the distance-to-boundary, hereafter. In the following analyses, we only consider when the mean SFA strength is smaller than the boundary, i.e., $\mu > 0$. Since when $\mu < 0$, the bump movement is in the travelling wave state for most of the time and is no longer a stochastic process. The second part, $\sigma_m/(2\sqrt{\pi}am)\xi_{m,0}(t)$, reflects the fluctuations of the SFA strength over time, and we denote $\gamma = \sigma_m/(2\sqrt{\pi}am)$ as the noise-to-strength ratio hereafter. Note that if the SFA strength is fixed, i.e., $\gamma = 0$, Eq. (15) degenerates into an Ornstein–Uhlenbeck (OU) process [37]. In such a case, the stationary distribution of $\mathbf{s}(t)$ has a Gaussian form, which leads to the Brownian motion of the bump position $\mathbf{z}(t)$.

Non-Brownian motion occurs when the SFA strength fluctuates, i.e., the noise-to-strength ratio $\gamma > 0$. In such a case, we can obtain the stationary distribution of $\mathbf{s}(t)$ by solving the corresponding Fokker-Planck equation (see SI.2 for the details), which gives,

$$p(s_i) = c_0 \left(\sigma_s^2 + \gamma^2 s_i^2\right)^{-(1+\mu/\gamma^2)}, \quad \text{for } i = \{1, 2\}, \tag{16}$$

where $s_1$ and $s_2$ represent the coordinates of $\mathbf{s}$ on the axes of $x_1$ and $x_2$, respectively. $\sigma_s^2 = \left(\sqrt{2}\tau_v \sigma_U/(\sqrt{\pi}\tau \tilde{A}_u)\right)^2 + \left(\sqrt{2}a\gamma\right)^2$ and $c_0$ is a normalization constant. From Eq. (14), the displacement of $\mathbf{z}(t)$ in a short time interval $\delta t$ is calculated to be $\|\Delta \mathbf{z}\| = \|m\mathbf{s}\delta t/\tau + \sqrt{2\delta t/(\pi\tau)}\sigma_U/\tilde{A}_u\xi_{U,1}\|$. Replacing $\mathbf{s}$ with its stationary distribution given by Eq. (16), we finally derive the distribution of $\|\Delta \mathbf{z}\|$, which is written as,

$$p(\|\Delta \mathbf{z}\|) \sim \|\Delta \mathbf{z}\|^{-1-(1+2\mu/\gamma^2)}, \tag{17}$$

which satisfies the power-law distribution (see SI.3 for the details). Comparing Eq. (17) with Eq. (1), we obtain the Lévy exponent of the bump movement,

$$\alpha = 1 + \frac{2\mu}{\gamma^2}. \tag{18}$$

It shows that the Lévy exponent is determined by two factors, which are the distance-to-boundary $\mu$ and the noise-to-strength ratio $\gamma$.

Based on the above theoretical analyses and comprehensive simulations, we investigate the effects of $\mu$ and $\gamma$ on the mobility of the bump state in the attractor space, and the results are summarized in Fig. 3. To elucidate the mechanism of generating Lévy flights clearly, we inspect the effects of $\mu$ and $\gamma$ separately by varying only one of them each time while fixing the other. The results are introduced below.

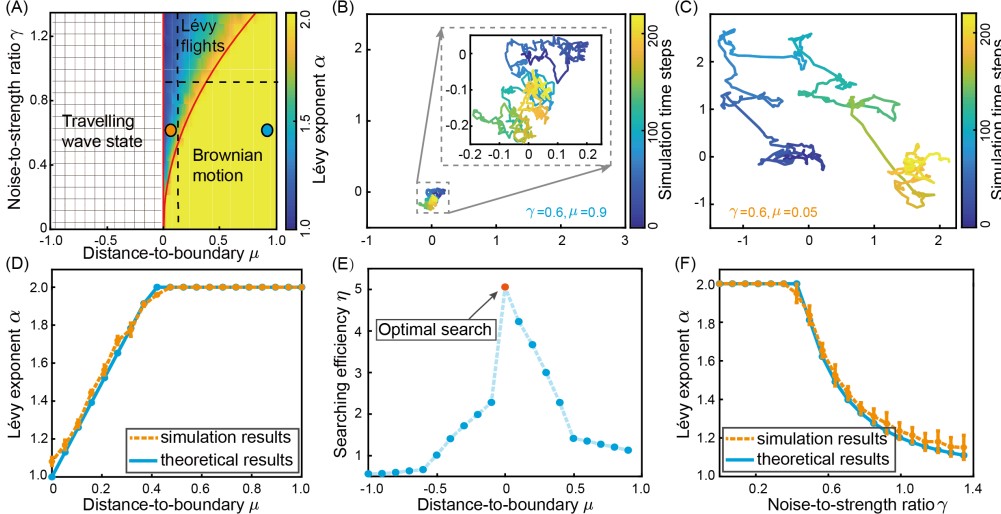

Figure 3: Lévy flights in a two-dimensional CANN. (A) The phase diagram of the network with respect to the distance-to-boundary $\mu$ and the noise-to-strength ratio $\gamma$. (B) An example of Brownian motion in the attractor space with $\mu = 0.95$ and $\gamma = 0.6$, corresponding to the blue point in (A). The inset is a close-up of the local Brownian motion. (C) An example of Lévy flights in the attractor space with $\mu = 0.05$ and $\gamma = 0.6$, corresponding to the orange point in (A). (D) The Lévy exponent $\alpha$ vs. $\mu$ with $\gamma = 0.9$, corresponding to the horizontal dashed line in (A). Note that when $\mu > \gamma^2/2$, i.e., $\mu > 0.405$, all $\alpha > 2$ (Brownian motion) will converge to $\alpha = 2$ due to the Central Limit Theorem. (E) The search efficiency $\eta$ vs. $\mu$. The optimal search is achieved when $\mu \to 0$, indicated by the orange point. (F) The Lévy exponent $\alpha$ vs. $\gamma$ with $\mu = 0.1$, corresponding to the vertical dashed line in (A). For the setting of other parameters, see SI.4 for the details.

**The effect of the distance-to-boundary** $\mu$. Fixing the noise-to-strength ratio $\gamma$ while varying $\mu$, we have the following observations: a) In the case of $\mu \geq \gamma^2/2$, which gives $\alpha \geq 2$, the bump movement displays Brownian motion (Fig. 3B). This is because when $\mu \geq \gamma^2/2$, the SFA strength is far away from the boundary, which is too weak to generate the travelling wave state. Thus, the bump movement is mainly driven by noise fluctuations in the neural dynamics (Eq. (2)). b) In the case of $0 < \mu < \gamma^2/2$, which gives $1 < \alpha < 2$, the bump movement displays Lévy flights (Fig. 3C). In this parameter regime, decreasing $\mu$ will decrease the Lévy exponent $\alpha$ from 2 to 1 gradually (Fig. 3D). This is because as $m$ gets closer to the travelling wave boundary, adaptation noises become more likely to push the SFA strength to cross the boundary, and hence long-jump movement occurs more frequent compared to local Brownian motion. When $m$ eventually approaches the travelling wave boundary, i.e., $\mu \to 0$, which gives $\alpha \to 1$, the network achieves the optimal Lévy search in the attractor space. Note that in the simulation, we can only estimate the value of $\alpha$ from a truncated power-law distribution, as very large jumps exceeding the size of the attractor space are excluded. This explains why the simulated Lévy exponent is slightly larger than the theoretical value of $\alpha = 1$ (see Fig. 3D). c) In the case of $\mu < 0$, the network dynamics will be dominated by the travelling wave state, which overrides the effects of both neuronal and adaptation noises, and the bump movement is no longer stochastic. d) We observe that the search efficiency of the bump (measured by the number of locations in the attractor space visited by the bump in a unit distance) reaches the optimal value when $\mu \to 0$ (Fig. 3E), manifesting the characteristics of Lévy flights.

**The effect of the noise-to-signal ratio $\gamma$.** Fixing the distance $\mu$ (e.g, $\mu = 0.1$, which is close to the travelling wave boundary) while varying $\gamma$, we have the following observations: a) In the case of $\gamma \leq \sqrt{2\mu}$, which gives $\alpha \geq 2$, the bump movement displays Brownian motion. This is understandable: although the mean SFA strength is close to the travelling wave boundary, the adaptation noises are too small to push the SFA strength to cross the boundary, and hence the bump movement is mainly driven by noise fluctuations in the neural dynamics (Eq. (2)). b) In the case of $\gamma > \sqrt{2\mu}$, which gives $1 < \alpha < 2$, the bump movement displays Lévy flights. Increasing $\gamma$ implies that adaptation noises have higher chances to push the SFA strength to cross the boundary, and hence the Lévy exponent $\alpha$ will keep decreasing, until it approaches the value of $\alpha = 1$ (Fig. 3F).

## 3 Lévy flights in free memory retrieval

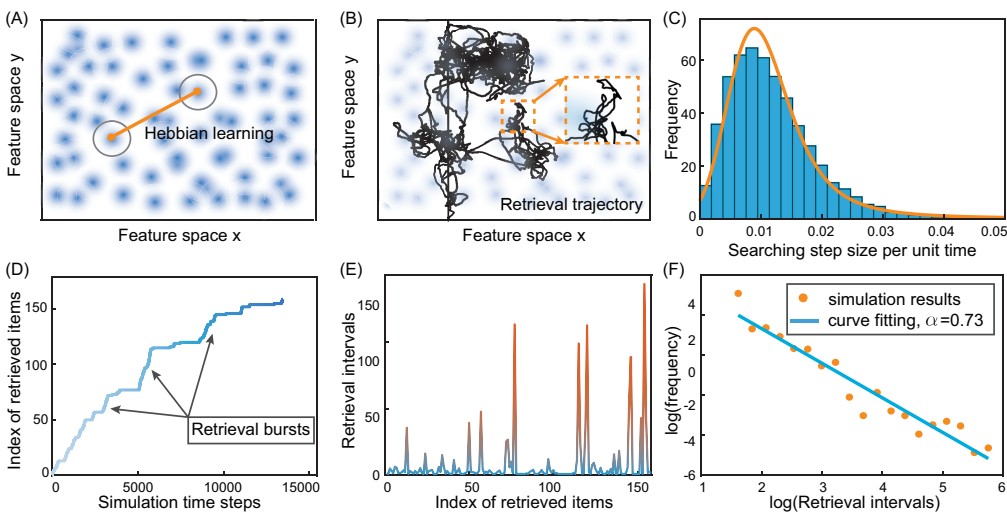

Figure 4: Modelling Lévy flights in free memory retrieval with an attractor network modulated by noisy adaptation. (A) Memory items are randomly distributed on the two-dimensional feature space and neuronal connections are determined by Hebbian learning. (B) An example of the retrieval trajectory in the feature space. (C) The histogram of step sizes of the retrieval trajectory in (B), which follows a power-law tailed distribution. (D) The index of retrieved items vs. the simulation time. Retrieved items burst intermittently over time. (E) Retrieval intervals vs. the index of retrieved items. Long intervals grow exponentially with interruptions by bursts of short retrieval intervals. (F) The log-log plot of the frequency distribution of retrieval intervals, with an estimated Lévy exponent $\alpha = 0.73$. For the setting of other parameters, see SI.4 for the details.

The first study of Lévy flights in human cognition was carried out by Rhodes and Turvey with a free memory retrieval task [11]. In the experiment, eight participants were asked to verbally recall animal names as many as possible (without repetition) within about 20 minutes, and the time intervals between successive recalls were recorded. Intriguingly, they found an exponential increase of long retrieval intervals that are interrupted by bursts of short intervals. Overall, the retrieval intervals can be described by Lévy flights with exponents of $0.37 \leq \alpha \leq 0.98$ for different participants. Here, we show that this Lévy-fight behavior of free memory retrieval can be described by random search on a semantic graph which is modelled by an attractor network with noisy adaptation.

Specifically, we build a Hopfield-like attractor network, in which each memory item is represented by a localized neural population randomly distributed in a two-dimensional feature space. The neuronal connections are determined by Hebbian learning, such that each memory item is encoded as an attractor of the network (Fig. 4A). Successful retrieval of a memory item means that the network state falls into the corresponding attractor. Note that in such a network, the stationary states no longer form a continuous manifold of neutral stability as in a CANN. Even so, adaptation implemented by SFA can still drive the network state to travel among different attractors if the adaptation strength is strong enough. Thus, when the mean and variance of the SFA strength are set properly, the network state

switches intermittently between local motion and long-jump motion in the feature space, exhibiting the characteristics of Lévy flights. This is confirmed by the simulation results in Fig. 4B&C.

We have shown that when the memory recall in the human brain is treated as a random search on a semantic graph on which memory items are uniformly distributed, the whole process can be described as Lévy flights in the attractor space of a Hopfield-like attractor network. To reproduce the experiment findings which show that the time intervals between successive recalls follow a long-tail power-law distribution, we further convert the Lévy flights in the 2-dimension space as shown in Fig. 4A into their counterparts in the temporal domain. For this purpose, we calculate the time intervals between successive retrievals which are measured by the duration of the network state travelling from one attractor to another (note that each attractor can only be visited once as required in the human experiment, and the recall is terminated once the network state visits the same attractor again due to local motion). Fig. 4D shows that the number of retrieved items scales logarithmically with the simulation time, and memory items are recalled in burst intermittently, which agrees well with the experimental finding (see Fig.1b in [11]). Further analysis shows that long retrieval intervals are interrupted by bursts of very short intervals and the length of long intervals increases exponentially over time (Fig. 4E), which agrees very well with the experimental finding (see Fig.1a in [11]). We also calculate the Lévy exponent by a linear fitting of the log-log frequency distribution of retrieval intervals and obtain $\alpha = 0.73$ (Fig. 4F), which is in the range of the experimental finding $0.37 \leq \alpha \leq 0.98$. It is also worth noting that this $\alpha$ shows the temporal property of Lévy flights (retrieval intervals), while the $\alpha$ (not shown) which can be obtained in Fig. 4C shows the spatial property of Lévy flights (step sizes on the graph).

## 4    Conclusion and Discussion

In the present study, we have studied a noisy-adaptation-modulated attractor network model to elucidate the mechanism of generating Lévy flights at the circuit level in neural systems. By analyzing the dynamics of a CANN with noisy SFA, we find that when the mean SFA strength is set close to the travelling wave boundary, noises in adaptation can easily cause the network state to intermittently switch between local motion and long-jump motion, displaying Lévy-flight patterns in the attractor space. We theoretically derive the power-law distribution of the step sizes of the bump movement, and show that the Lévy exponent is determined by the joint effect of two key factors, i.e., the distance of the mean SFA strength to the travelling wave boundary, and the noise level in the adaptation. We also demonstrate that optimal search in the attractor space is achieved when the mean SFA strength approaches the travelling wave boundary. Simulation results agree with our theoretical analyses very well.

Furthermore, we generalize the theoretical findings to a Hopfield-like attractor network and use them to explain the Lévy-flight patterns found in free memory retrieval. The Hopfield-like attractor network encompasses a low-dimensional attractor space which defines a semantic graph of memory items, and movement of the network state in the attractor space corresponds to a random mental exploration on the semantic graph. By introducing noisy adaptation into the neural dynamics, we observe that the network exhibits Lévy-flight moving patterns in the attractor space, which reproduce the experimental findings very well, including bursts of retrieved items during the retrieval process, the exponential increase of long retrieval intervals over time, and the value of the Lévy exponent.

**Related works of generating Lévy flights in the brain.**    Mathematical models for demonstrating Lévy flights in movement patterns of living things have been intensively studied in the literature, ranging from sub-cellular [38, 39], to the individual organism [5, 7], and to population scales [40, 41]. Typically, these studies built up phenomenological models which describe the time evolution of the diffusion and transport of a moving object, e.g., by coupled continuous time random walks (CTRW) [42, 18], by various fractional PDEs [43, 44], and by integro-differential equations [45, 46]. Up to now, mechanisms of generating Lévy flights in the brain (especially at the neural circuit level), which underlie the generation of macroscopic Lévy-flight behaviors across different levels of living organisms, have been rarely studied. Very recently, two works have drawn our attention. First is the work from McNamee et al. [47], in which they proposed an entorhinal-hippocampal network model with linear feedback that implements spectral modulation of sequence generation. Their model can explain the Lévy-flight patterns found in foraging behaviors and other cognitive functions in animals. However, it is not a circuit model based on detailed neural dynamics, but instead based on a master

equation that describes the time evolution of the state distribution in the network. In our work, we start from a detailed continuous attractor network model with generic neural features. We derive the dynamics of the network state in the attractor space and carry out theoretical analysis to elucidate how neural adaptation causes Lévy flights in the network. These detailed analyses enables us to show the relationships between the critical parameters in our model (i.e., the distance-to-boundary $\mu$ and the noise-to-signal ratio $\gamma$) and the properties of Lévy flights quantitatively. We expect that this detailed computational model can give us more insight into understanding the generation of Lévy flights in the brain and guide us to carry out further experimental and theoretical studies.

The second related work is from Gong and his co-authors [48, 49, 50]. In the study [48], Wardak and Gong proposed a leaky integrate and fire (LIF) neuron model with heterogeneous connection strengths, and demonstrated that the membrane potential of the LIF neuron undergoes Lévy flights. They carried out mathematical analysis to show that the total synaptic input to the LIF neuron has nonequilibrium Lévy noises when the distribution of the heterogeneous synaptic connections has a power-law tail, and thus the membrane potential probability density satisfies a fractional Fokker-Planck equation. Liu et al. further extend the model to a balanced spiking neural circuit model to explain the Lévy-flight patterns observed in the gamma bursts in the area MT of marmoset monkeys [50]. Notably, they also found that the Lévy-flight patterns observed in their model arise from the critical transitions between the asynchronous and propagating wave states, similar to the transitions between the localized bump and travelling wave states in our model. Although both their and our studies observed the Lévy-flight phenomenon in neural networks, the models and the associated mechanisms for generating Lévy flights are very different. Specifically, their model considers power-law tailed synaptic connections, which lead to Lévy noises in a LIF neuron. Theoretical analyses showed that the membrane potential can then be described by a fractional Fokker-Planck equation with Lévy noises, which is a well-known mathematical strategy to generate Lévy flights [51, 52, 53]. On the other hand, we consider an attractor neural network with noisy adaptation, and the network state movement is described by a Fokker-Planck equation with multiplicative Gaussian noises, which belongs to another mathematical strategy to generate Lévy flights [54, 55]. It will be interesting to explore in future works whether one mechanism is more biologically plausible or whether both mechanisms exist in the brain but are applied in different situations.

**Other potential mechanisms of generating Lévy flights in attractor models.**    The mechanism we propose exploits noisy adaptation to generate Lévy flights. Both noises and adaptation are the intrinsic properties ubiquitously observed in neural systems, and hence they provide an internal strategy to generate Lévy flights without relying on external inputs. This internal-based mechanism is appealing for efficient information searching in an unknown environment with scarce targets, e.g., for animal foraging in an open space with sparse prey. Notably, recent research has revealed that when targets are abundantly distributed in the environment, animals can switch the search strategy from Lévy flights to Brownian-motion [41, 56]. Our model can also account for this strategy switch phenomenon. As shown in Fig. 3A, by varying the SFA strength, which could be realized by neural modulation in reality, the network dynamics can switch between Brownian motion and Lévy flights. In addition to SFA, neural systems have other mechanisms to implement adaptive responses, including, for instance, the short-term depression (STD) of synapses between neurons [57] and the long-range feedback inhibition between cortical areas [58]. These mechanisms can also destabilize the attractor states of a neural circuit via negative feedback, and generate Lévy flights as analyzed in this study. The differences in these mechanisms are that they operate at different spatial and temporal scales. For instance, SFA occurs at the single neuron level and has a time scale of less than one hundred milliseconds, STD occurs at the synapse level and has a time scale of hundreds to thousands milliseconds, and the long-range feedback inhibition occurs between brain regions and has a time scale of tens of milliseconds. Thus, the brain may employ different strategies to generate Lévy-flight patterns at different temporal scales and in different brain regions, and they support efficient information processing in different cognitive functions. It will be interesting to explore these issues in our future work.

## Acknowledgement

This work was supported by Beijing Academy of Artificial Intelligence (BAAI), Guangdong Province with grant (No. 2018B030338001, S. Wu), Huawei Technology Co.,Ltd (No. YBN2019105137 S. Wu) and the National Natural Science Foundation of China (No. 4861425025, T.J.Huang).

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
