# Noisy Adaptation Generates Lévy Flights in Attractor Neural Networks
## Supplementary Information

## 1 The dynamics of the bump heights and positions

In this section, using the projection method, we derive the dynamics of the bump heights $A_u(t)$ and $A_v(t)$, and the dynamics of the bump positions $\mathbf{z}(t)$ and $\mathbf{s}(t)$.

### 1.1 The dynamics of $A_u$ and $A_v$

The dynamics of the CANN with SFA is expressed as,

$$\tau \frac{\partial U(\mathbf{x},t)}{\partial t} = -U(\mathbf{x},t) + \rho \int_{\mathbf{x}'} J(\mathbf{x},\mathbf{x}') r(\mathbf{x}',t) \, \mathrm{d}\mathbf{x}' - V(\mathbf{x},t) + \sigma_U \xi_U(\mathbf{x},t), \quad (1)$$

$$\tau_v \frac{\partial V(\mathbf{x},t)}{\partial t} = -V(\mathbf{x},t) + [m + \sigma_m \xi_m(\mathbf{x},t)] U(\mathbf{x},t), \quad (2)$$

$$r(\mathbf{x},t) = \frac{U^2(\mathbf{x},t)}{1 + k\rho \int_{\mathbf{x}'} U^2(\mathbf{x}',t) \, \mathrm{d}\mathbf{x}'}. \quad (3)$$

As stated in the main text, the presumed network state have the following form,

$$U(\mathbf{x},t) = A_u(t) \exp\left\{ -\frac{[\mathbf{x} - \mathbf{z}(t)]^2}{4a^2} \right\}, \quad (4)$$

$$r(\mathbf{x},t) = A_r(t) \exp\left\{ -\frac{[\mathbf{x} - \mathbf{z}(t)]^2}{2a^2} \right\}, \quad (5)$$

$$V(\mathbf{x},t) = A_v(t) \exp\left\{ -\frac{[\mathbf{x} - (\mathbf{z}(t) - \mathbf{s}(t))]^2}{4a^2} \right\}. \quad (6)$$

The dynamics of the CANN is dominated by a few motion modes [1]. This motivates us to simplify the analysis by projecting the network dynamics onto these motion modes. In the current study, we adopt the first two dominating motion modes, with one the bump height mode and the other the bump position mode, which are,

$$u_0(\mathbf{x}|\mathbf{z}) = \frac{1}{a\sqrt{2\pi}} \exp\left\{ -\frac{[\mathbf{x} - \mathbf{z}(t)]^2}{4a^2} \right\}, \quad (7)$$

$$u_1(\mathbf{x}|\mathbf{z}) = \frac{1}{a^2\sqrt{2\pi}} [\mathbf{x} - \mathbf{z}(t)] \exp\left\{ -\frac{[\mathbf{x} - \mathbf{z}(t)]^2}{4a^2} \right\}. \quad (8)$$

In the following, we will mainly focus on the derivation of the dynamics of $A_u$, as the derivation of $A_v$ is similar. Before carrying out the projection method, we first substitute the presumed network state (Eq. (4-6)) into the CANN dynamics (Eq. (1)), which gives the expression of $A_u$ (to prevent the

35th Conference on Neural Information Processing Systems (NeurIPS 2021), Sydney, Australia.

wrapping of expressions, we show the results of left side and right side respectively),

$$l-side = \tau\exp\left[-\frac{(\mathbf{x}-\mathbf{z})^2}{4a^2}\right]\frac{\mathrm{d}A_u}{\mathrm{d}t}+\tau\frac{A_u(\mathbf{x}-\mathbf{z})}{2a^2}\exp\left[-\frac{(\mathbf{x}-\mathbf{z})^2}{4a^2}\right]\frac{\mathrm{d}\mathbf{z}}{\mathrm{d}t}, \tag{9}$$

$$r-side = \left(-A_u+\frac{\rho J_0 A_r}{2}\right)\exp\left[-\frac{(\mathbf{x}-\mathbf{z})^2}{4a^2}\right]-A_v\exp\left[-\frac{(\mathbf{x}-\mathbf{z}+\mathbf{s})^2}{4a^2}\right]$$
$$+\sigma_U\xi_U(\mathbf{x},t). \tag{10}$$

As mentioned in the main text, projecting a function $f(\mathbf{x})$ onto a mode $u(\mathbf{x})$ equals to computing $\int_{\mathbf{x}}f(\mathbf{x})u(\mathbf{x})d\mathbf{x}/\int_{\mathbf{x}}u^2(\mathbf{x})d\mathbf{x}$. Therefore, by projecting both sides onto the motion mode $u_0(\mathbf{x}|\mathbf{z})$, we obtain,

$$l-side = \tau 2\pi a^2\frac{\mathrm{d}A_u}{\mathrm{d}t}+\frac{\tau A_u}{2a^2}\frac{\mathrm{d}\mathbf{z}}{\mathrm{d}t}\int_{-\infty}^{\infty}\mathrm{d}\mathbf{x}(\mathbf{x}-\mathbf{z})\exp\left[-\frac{(\mathbf{x}-\mathbf{z})^2}{2a^2}\right], \tag{11}$$

$$r-side = \left(-A_u+\frac{\rho J_0}{2}A_r\right)2\pi a^2-A_v\exp\left[-\frac{\mathbf{s}^2}{8a^2}\right]2\pi a^2+\sqrt{2\pi}a\sigma_U\xi_{U,0}. \tag{12}$$

Note that the part inside the integral in Eq. (11) is an odd function which integrates to zero. Intuitively, this odd function can be regard as the variation of the bump position along the dimension of the bump height, which are orthogonal to each other. In other words, this term has no effect on the dynamics of $A_u$. Thus, by equating both sides, we obtain,

$$\tau\frac{\mathrm{d}A_u}{\mathrm{d}t}=\left(-A_u+\frac{\rho J_0}{2}A_r\right)-A_v\exp\left[-\frac{\mathbf{s}^2}{8a^2}\right]+\frac{1}{\sqrt{2\pi}a}\sigma_U\xi_{U,0}, \tag{13}$$

where $\xi_{U,0}$ is obtained by projecting the noise term $\xi_U$ onto the motion mode $u_0(\mathbf{x}|\mathbf{z})$. More concretely, the newly defined noise term is given by,

$$\xi_{U,0}(t)=\int_{\mathbf{x}}\xi_U(\mathbf{x},t)u_0(\mathbf{x}|\mathbf{z})\mathrm{d}\mathbf{x}. \tag{14}$$

It is straightforward to check that

$$\langle\xi_{U,0}(t)\rangle = \int_{\mathbf{x}}\langle\xi_U(\mathbf{x},t)\rangle u_0(\mathbf{x}|\mathbf{z})\mathrm{d}\mathbf{x}=0, \tag{15}$$

$$\langle\xi_{U,0}(t)\xi_{U,0}(t')\rangle = \int_{\mathbf{x}}\mathrm{d}\mathbf{x}\int_{\mathbf{x}'}\mathrm{d}\mathbf{x}'\langle\xi_U(\mathbf{x},t)\xi_U(\mathbf{x}',t')\rangle u_0(\mathbf{x}|\mathbf{z})u_0(\mathbf{x}|\mathbf{z})$$
$$= \delta(t-t'), \tag{16}$$

with $\delta$ is a Dirac delta function. Eqs. (15&16) imply that the newly defined noise term $\xi_{U,0}$ is still a Gaussian white noise with zero mean and unit variance, as stated in the main text.

Furthermore, the previous work [2] has shown that, without noises, the separation $\mathbf{s}(t)$ between $U(x,t)$ and $V(x,t)$ has a stationary solution, which is expressed as $\mathbf{s}^2=a^2(1-\sqrt{m/(\tau/\tau_v)})$. This implies that, as long as the SFA strength is set around the travelling wave boundary $\tau/\tau_v$, then $\mathbf{s}^2\ll a^2$ always holds (Fig. 1A), which means that $\mathbf{s}^2/a^2\approx 0$. Therefore, Eq. (13) can be further simplified as,

$$\tau\frac{\mathrm{d}A_u}{\mathrm{d}t}=-A_u-A_v+\frac{J_0\rho A_r}{2}+\frac{\sigma_U}{a\sqrt{2\pi}}\xi_{U,0}(t), \tag{17}$$

which corresponds to Eq. (10) in the main text.

To obtain the dynamics of $A_v$, we substitute the presumed network state (Eq. (4-6)) into the SFA dynamics (Eq. (2)), which gives,

$$l-side = \tau_v\exp\left[\frac{(\mathbf{x}-\mathbf{z}+\mathbf{s})^2}{4a^2}\right]\frac{\mathrm{d}A_v}{\mathrm{d}t}+\tau_v\frac{A_v(\mathbf{z}-\mathbf{x}-\mathbf{s})}{2a^2}\exp\left[\frac{(\mathbf{x}-\mathbf{z}+\mathbf{s})^2}{4a^2}\right]\frac{\mathrm{d}\mathbf{z}-\mathrm{d}\mathbf{s}}{\mathrm{d}t}, \tag{18}$$

$$r-side = -\tau_v A_v\exp\left[\frac{(\mathbf{x}-\mathbf{z}+\mathbf{s})^2}{4a^2}\right]+[m+\sigma_m\xi_m(\mathbf{x},t)]A_u\exp\left[\frac{(\mathbf{x}-\mathbf{z})^2}{4a^2}\right]. \tag{19}$$

Similarly, we project both sides onto the motion mode $u_0(\mathbf{x}|\mathbf{z})$, and obtain the dynamics of $A_v$,

$$\tau_v\frac{\mathrm{d}A_v}{\mathrm{d}t}=-A_v+mA_u+\frac{\sigma_m A_u}{2a\sqrt{\pi}}\xi_{m,0}(t), \tag{20}$$

which corresponds to Eq. (11) in the main text. Here, $\xi_{m,0}$ is also a Gaussian white noise with zero mean and unit variance.

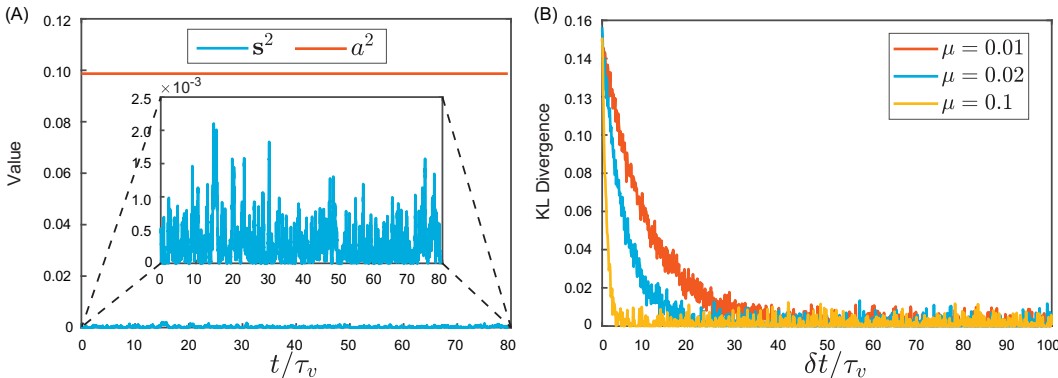

Figure 1: (A) Simulation results confirm that the amplitude of $s$ is much smaller than $a$. (B) The KL divergences between the conditional distribution $p(s(t_0 + \delta t)|s(t_0))$ and the stationary distribution $p^{st}(s)$ with different values of $\mu$. They decay exponentially to the stationary values with a rate of $O\big(\exp(-\mu\delta t/\tau_v)\big)$.

## 1.2 The dynamics of $\mathbf{z}(t)$ and $\mathbf{s}(t)$

We only present the derivation of the dynamics of $\mathbf{z}(t)$, as the case for $\mathbf{s}(t)$ is similar.

Projecting both sides the network dynamics (Eq. (9&10)) onto the motion mode $u_1(\mathbf{x}|\mathbf{z})$, we obtain,

$$l - side = \tau\pi a^2 A_u \frac{d\mathbf{z}}{dt} + \tau\frac{dA_u}{dt}\int_{-\infty}^{\infty} d\mathbf{x}(\mathbf{x} - \mathbf{z})\exp\left[-\frac{(\mathbf{x} - \mathbf{z})^2}{2a^2}\right], \tag{21}$$

$$r - side = \pi a^2 A_v \mathbf{s}\exp\left[-\frac{\mathbf{s}^2}{8a^2}\right] + a^2\sqrt{2\pi}\sigma_U\xi_{U,1}(t). \tag{22}$$

Note that the part inside the integral in Eq. (21) is also an odd function which integrates to zero. Thus, by equating both sides and utilizing the fact that $\mathbf{s}^2/a^2 \approx 0$, we obtain,

$$\tau\frac{d\mathbf{z}}{dt} = \frac{A_v}{A_u}\mathbf{s} + \sqrt{\frac{2}{\pi}}\frac{\sigma_U}{A_u}\xi_{U,1}(t), \tag{23}$$

which corresponds to Eq. (12) in the main text. Here, $\xi_{U,1}$ is obtained by projecting the noise term $\xi_U$ to the motion mode $u_1(\mathbf{x}|\mathbf{z})$, i.e.,

$$\xi_{U,1}(t) = \int_{\mathbf{x}} \xi_U(\mathbf{x}, t)u_1(\mathbf{x}|\mathbf{z})d\mathbf{x}. \tag{24}$$

It is straightforward to check that

$$\langle\xi_{U,1}(t)\rangle = \int_{\mathbf{x}}\langle\xi_U(\mathbf{x}, t)\rangle u_0(\mathbf{x}|\mathbf{z})d\mathbf{x} = 0, \tag{25}$$

$$\langle\xi_{U,1}(t)\xi_{U,1}(t')\rangle = \int_{\mathbf{x}}d\mathbf{x}\int_{\mathbf{x}'}d\mathbf{x}'\langle\xi_U(\mathbf{x}, t)\xi_U(\mathbf{x}', t')\rangle u_1(\mathbf{x}|\mathbf{z})u_1(\mathbf{x}|\mathbf{z})$$

$$= \delta(t - t'). \tag{26}$$

This indicates that the newly defined noise term $\xi_{U,1}$ is still a Gaussian white noise with zero mean and unit variance, as stated in the main text.

Similarly, we can obtain the dynamics of $\mathbf{s}(t)$ by projecting Eq.(18-19) onto $u_1(\mathbf{x}|\mathbf{z})$, which gives,

$$\tau_v\frac{d\mathbf{s}}{dt} = \left(\frac{\tau_v A_v}{\tau A_u} - \frac{mA_u}{A_v} - \frac{\sigma_m A_u \xi_{m,0}}{2A_v\sqrt{\pi}a}\right)\mathbf{s} + \frac{\tau_v\sigma_U}{\tau A_u}\sqrt{\frac{2}{\pi}}\xi_{U,1} - \frac{\sigma_m A_u}{A_v}\sqrt{\frac{1}{2\pi}}\xi_{m,1}. \tag{27}$$

This corresponds to Eq. (13) in the main text. Here, $\xi_{m,1}$ is also a Gaussian white noise with zero mean and unit variance.

## 2 The stationary distribution of s(t)

In this section, we present the detailed derivation of the stationary distribution of $\mathbf{s}(t)$.

First of all, we re-write the dynamics of $\mathbf{s}(t)$ (see Eq. 15 in the main text) as follows,

$$\tau_v \frac{\mathrm{d}s_i}{\mathrm{d}t} = -(\mu + \gamma \xi_m)s_i + \sigma_s \xi_{s_i}, \quad i = \{1, 2\}. \tag{28}$$

Here, $\mu = 1 - m\tau_v/\tau$ is the distance-to-boundary, and $\sigma_m/(2\sqrt{\pi}am)\xi_{m,0}(t)$ is the noise-to-strength ratio, as defined in the main text. The two variables $s_1$ and $s_2$ are the components of $\mathbf{s}$ in the two-dimensional space, respectively. Without loss of generality, we only study one component, and for clearance, we denote $s_i(t)$ and $\xi_{s_i}(t)$ as $s(t)$ and $\xi_s(t)$, respectively.

When the noise-to-strength ratio $\gamma = 0$, the dynamics of $\mathbf{s}(t)$ the Ornstein–Uhlenbeck (OU) process [3]. The stationary distribution of $s$ is solved to be,

$$p^{st}(s) = \sqrt{\frac{\mu}{\pi \sigma_s^2}} exp\left[-\frac{\mu s^2}{\sigma_s^2}\right]. \tag{29}$$

When $\gamma > 0$, the drift term in the dynamics of $\mathbf{s}(t)$ is affected by the multiplicative noise $\xi_m$. Utilizing Itô calculus, we can rewrite Eq.(28) as a first order difference equation,

$$s(t + \mathrm{d}t) = s(t) + \int_t^{t+\mathrm{d}t} \left(-\frac{\mu s(t')}{\tau_v} + \frac{\gamma s(t')}{\sqrt{\tau_v}}\xi_m(t') + \frac{\sigma_s}{\sqrt{\tau_v}}\xi_s(t')\right)\mathrm{d}t', \tag{30}$$

$$= s(t) - \frac{\mu s(t)}{\tau_v}\mathrm{d}t + \frac{1}{\sqrt{\tau_v}}\left(-s(t)\gamma \mathrm{d}t\overline{\xi_m} + \sigma_s \mathrm{d}t\overline{\xi_s}\right), \tag{31}$$

where $\mathrm{d}t\overline{\mu}$ and $\mathrm{d}t\overline{\xi_s}$ are the Ito prescriptions in the limit of $\mathrm{d}t \to 0$.

To derive the Fokker-Planck equation, we adopt a smooth trial function $R(s)$ proposed by Rivers [4], and compute its average value at time $t$, which is expressed as,

$$\langle\langle R(t) \rangle\rangle = \int R(s)p(s, t)\mathrm{d}s, \tag{32}$$

where $p(s, t)$ is the distribution of $s(t)$ at time $t$. Consider the evolution of the average value of $R(t)$ in a short interval $dt$ at time $t$, which is given by,

$$\langle\langle R(t + \mathrm{d}t) \rangle\rangle = \left\langle \int R\left(s - \frac{\mu s}{\tau_v}\mathrm{d}t + \frac{1}{\sqrt{\tau_v}}\left(-s\gamma \mathrm{d}t\overline{\xi_m} + \sigma_s \mathrm{d}t\overline{\xi_s}\right)\right)p(s, t)\mathrm{d}s \right\rangle. \tag{33}$$

With the first-order Taylor-series expansion of the trial function $R(\cdot)$ at $s$, the right side of Eq.(33) is expressed as,

$$\left\langle \int ds\, p(s,t)R\left(s - \frac{\mu s}{\tau_v}\mathrm{d}t + \frac{1}{\sqrt{\tau_v}}(-s\gamma \mathrm{d}t\overline{\xi_m} + \sigma_s \mathrm{d}t\overline{\xi_s})\right)\right\rangle =$$
$$\left\langle \int ds\, p(s,t)\left[R(s) + \mathrm{d}tR'(s)\left(-\frac{\mu}{\tau_v}s\right) + \mathrm{d}tR''(s)\left(\frac{\sigma_s^2 + \gamma^2 s^2}{2\tau_v}\right)\right]\right\rangle. \tag{34}$$

Note that the left side of Eq.(33) corresponds to the partial derivative of $p(s, t)$ with respect to $t$, and the right side of Eq.(34) corresponds to the partial derivative of $p(s, t)$ with respect to $s$. Thus, we can achieve the following Fokker-Planck expression of the distribution of $s(t)$,

$$\frac{\partial p(s, t)}{\partial t} = -\frac{\partial}{\partial s}\left(-\frac{\mu}{\tau_v}sp(s, t)\right) + \frac{\partial^2}{\partial s^2}\left(\frac{\sigma_s^2 + \gamma^2 s^2}{2\tau_v}p(s, t)\right). \tag{35}$$

It describes the time evolution of the distribution of $s(t)$. The stationary distribution of $p^{st}(s)$ is achieved when

$$-\frac{\mu}{\tau_v}sp^{st}(s) = \frac{\mathrm{d}}{\mathrm{d}s}\left(\frac{\sigma_s^2 + \gamma^2 s^2}{2\tau_v}p^{st}(s)\right). \tag{36}$$

Thus, we obtain the stationary solution $p^{st}(s)$ as,

$$p^{st}(s) = c_0\left(\sigma_s^2 + \gamma^2 s^2\right)^{-(1+\mu/\gamma^2)}, \tag{37}$$

where $c_0$ is a normalization constant. This corresponds to Eq. (16) in the main text.

# 3 The distribution of $\|\Delta z\|$

In this section, we derive the distribution of $\|\Delta z\|$ based on the stationary distribution of $\mathbf{s}(t)$.

Recall that the simplified dynamics of $\mathbf{z}(t)$ and $\mathbf{s}(t)$ are expressed as,

$$\tau \frac{\mathrm{d}z}{\mathrm{d}t} = ms + \frac{\sigma_U}{\langle A_u \rangle} \sqrt{\frac{2}{\pi}} \xi_{U,1}(t), \tag{38}$$

$$\tau_v \frac{\mathrm{d}s}{\mathrm{d}t} = -(\mu + \gamma \xi_m)s + \sigma_s \xi_s, \tag{39}$$

where $\mu = 1 - m\tau_v/\tau$ is the distance-to-boundary, and $\sigma_m / (2\sqrt{\pi} am) \xi_{m,0}(t)$ is the noise-to-strength ratio, as defined in the main text. For simplicity and clearance, we consider $z$ and $s$ as the 1D components of the original two-dimensional variables $\mathbf{z}(t)$ and $\mathbf{s}(t)$. We also only consider the case of noisy SFA, i.e., $\gamma \neq 0$, which is necessary to generate Lévy flights in the attractor space, as stated in the main text.

According to the general Fokker-Planck expression of the distribution of $s(t)$ (Eq. (35)), we have,

$$\langle s(t + \delta t) \rangle = \langle s(t) \rangle \exp\left[-\frac{\mu \delta t}{\tau_v}\right], \tag{40}$$

$$\langle s(t + \delta t)s(t) \rangle - \langle s(t + \delta t) \rangle \langle s(t) \rangle = \left(\langle s^2(t) \rangle - \langle s(t) \rangle^2\right) \exp\left[-\frac{\mu \delta t}{\tau_v}\right]. \tag{41}$$

The exponential terms on the right side in the above equations indicate that $s(t)$ relaxes exponentially fast to the stationary distribution Eq. (37) on the time scale of $\delta t \gg \tau_v$, which is reflected by the KL divergence between the conditional distribution $p\left(s(t + \delta t)|s(t)\right)$ and the stationary distribution $p^{st}(s)$ (Fig. 1B). Lubashevsky et al. [5] demonstrated that the displacement of $z(t)$ in $\delta t$ (denote as $\Delta z$ hereafter) is dominated by the maximal value of $s$ in this time interval (denote as $s_{max}$ hereafter), that is, $\Delta z(\tau_v) \sim s_{max}\tau_v$. Due to the temporal independence of $s_{max}(t)$ when $\delta t \gg \tau_v$, the value of $\Delta z$ can be acquired by sampling from the stationary distribution of $s_{max}(t)$, i.e. the stationary distribution of $s(t)$. Thus, substituting Eq.(37) into Eq.(38), we can obtain the distribution of $\Delta z$,

$$p(\Delta z) \sim \Delta z^{-1+(1+2\mu/\gamma^2)}. \tag{42}$$

Up to now, we have obtained the distribution of $\Delta z$ as a single component of $\mathbf{z}$. To get the distribution of $\|\Delta z\|$, it is necessary to investigate the correlation between two components of $\mathbf{z}$ and $\mathbf{s}$. The correlation between $s_1$ and $s_2$ when both of them reach the stationary distribution is calculated to be,

$$\langle s_1(t_0 + \mathrm{d}t)s_2(t_0 + \mathrm{d}t) \rangle = \left[(1 - \frac{\mu}{\tau_v})^2 + \frac{\gamma^2 \mathrm{d}t}{\tau_v}\right] \langle s_1(t_0)s_2(t_0) \rangle. \tag{43}$$

In the case of $\mu > 0$ and $\gamma$ an infinitesimal, we have $\left[(1 - \mu/\tau_v)^2 + \gamma^2 \mathrm{d}t/\tau_v\right] < 1$, which indicates that the correlation between $s_1$ and $s_2$ will converge to 0 exponentially, i.e., the correlation between these two components can be ignored under the condition of stationary distribution. Since $\Delta z(\tau_v) \sim s_{max}\tau_v$, it is straightforward that the correlation between $\Delta z_1$ and $\Delta z_2$ can be ignored as well. Hence, $\|\Delta z\|$ (with $\|\Delta z\| = \sqrt{\Delta z_1^2 + \Delta z_2^2}$) will follow the same power-law distribution as $\Delta z$ (Eq. (42)), i.e.,

$$p(\|\Delta z\|) \sim \|\Delta z\|^{-1+(1+2\mu/\gamma^2)}. \tag{44}$$

This corresponds to Eq.(17) in the main text.

# 4 Hyper-parameter settings in simulations

Table 1 shows the hyper-parameters used in Fig. 3 in the main text.

Table 2 shows the hyper-parameters used in Fig. 4 in the main text.

Table 1: The hyper-parameters used in Fig. 3 in the main text.

| Hyper-parameters | Value |
| --- | --- |
| Time constants of $U$: $\tau$ | 1 |
| Time constants of $V$: $\tau_v$ | 100 |
| Neuron density: $\rho$ | $4096/\pi^2$ |
| Global inhibition strength: $k$ | 0.05 |
| Recurrent connection strength: $J_0$ | 1 |
| Recurrent connection radius: $a$ | $\pi/10$ |

Table 2: The hyper-parameters used in Fig. 4 in the main text.

| Hyper-parameters | Value |
| --- | --- |
| Neuron number: $N$ | 14400 |
| Number of encoded patterns: $n$ | 1600 |
| Time constants of U: $\tau$ | 10 |
| Time constants of V: $\tau_v$ | 25 |
| Recurrent connection range: $a$ | $\pi/10$ |
| Neuron density: $\rho$ | 364.75 |
| Global inhibition strength: $k$ | 0.05 |
| Recurrent connection strength: $J_0$ | 10 |
| Mean value of SFA strength: $\bar{m}$ | 0.56 |
| Variance of SFA strength: $\sigma_m$ | 6.4 |
| Input noise: $\sigma_U$ | 0.01 |
| Detection range: $r_c$ | 0.0785 |