# OpenReview forum: "Noisy Adaptation Generates Lévy Flights in Attractor Neural Networks"
_NeurIPS.cc/2021/Conference — NeurIPS 2021 Poster_

### Official Review · Reviewer_FVLu · 2021-07-15

**Rating:** 7
**Confidence:** 3

**Summary:**

The study describes a model of wave propagation in recurrent neural network displaced on a 2D plan. The neural activation is modeled with a simple divise inhibition model, and the lateral connectivity is given by a Gaussian kernel.  The paper reviews some works that have studied the stability of the activation pattern in this setting and the motion if the neuron model has spike frequency adaptation (SFA). It seems that this novelty here is to include noisy in the dynamics of adaptation. By solving this analytically, the authors shows Levy flight dynamics.

By studying a special case where the recurrent connectivity is derived from Hebbian learning, it is argued that it could model jumpy wandering thoughts.

**Limitations And Societal Impact:**

One limitation is that the Levy-Flights seem to appear only in the model when the adaptation is noisy. This seems to be a strong requirement because the chosen model of adaptation is simplified and I am convinced whether it captures well the wide range of adaptation mechanisms and their variety. I am not sure for instance that this model of adaptation is as solidly grounded on data as divisive inhibition. So would Levy flights emerge if the adaptation model is slightly different? or noisy in a different way? Perhaps at least in simulation it is possible to explore whether it is sufficient that noisy connectivity pattern or neuron position is enough to produce the effect rather that this particular model of noisy adaptation? I also wonder whether other types of frequency adaptation (for instance non-linear "spike" frequency adaptation involving r rather than U) would still lead to this behavior.

A second limitation is that the authors make an implicit far-fetched connection between Levy flights in behavior from Drosophilia and humans and Levy-flights in neural activity (line 35 to 44). The geographical motion of human is certainly explained by a different phenomena than the behavior Drosophilia, is it therefore ok to suggest that your model is an answer to all this by mentioning it in the intro? I believe it would be more convincing  to minimize or clarify this paragraph, at least I do not think that the presented model explains all these behavior. I do find that the speculation that the model could explain mind wandering is charming, although quite hypothetical. Ideally, this third section should acknowledge that this connection is a speculation or justify it better.

**Main Review:**

Overall the paper is very well written. The review of previous works is very clear even for me although I did not know about CANNs.

Many of the intermediate steps of the mathematical demonstration are inherited from other papers and the rest is detailed in the appendix. I tried to review some of that and I did not find problems but I did not go through the entire supplementary materials.

**Time Spent Reviewing:**

4 hours

---

> ### Author Response · Authors · 2021-08-10
> **Response to Reviewer FVLu**
>
> **We acknowledge the valuable comments of the reviewer.**
>
> To better address the concerns of the reviewer, we would like to first briefly summarize our model and its underlying mechanism for generating Levy flights. We first theoretically analyzed a neural circuit model, i.e., a continuous attractor neural network (CANN) with noisy spike frequency adaptation (SFA). Without noises in SFA, the network can only exhibit either Brownian motion or travelling wave, depending on the strength of SFA. When noises are included in SFA, the fluctuated SFA strengths can cause the network state to intermittently switch between Brownian motion and long-jump motion, resulting in Levy flights. Thus, in our model, the key element is a neural mechanism that can induce intermittent switches between local and long-jump motions of the network state. We then extended the study to a Hopfield-like net and observed the same Levy flight phenomenon reproducing the free memory recall experiment. This is understandable, as noise adaptation (SFA) can similarly induce intermittent switches between local motions and long-jump motions in the Hopfield-like net. Notably, both attractor networks and noisy SFA are biologically plausible.
>
> **In the below, we would like to address the concerns of the reviewer in detail.**
>
> **[Limitation 1]** ***One limitation is that the Levy-Flights seem to appear only in the model when the adaptation is noisy. This seems to be a strong requirement because the chosen model of adaptation is simplified and I am convinced whether it captures well the wide range of adaptation mechanisms and their variety. I am not sure for instance that this model of adaptation is as solidly grounded on data as divisive inhibition. So would Levy flights emerge if the adaptation model is slightly different? or noisy in a different way? Perhaps at least in simulation it is possible to explore whether it is sufficient that noisy connectivity pattern or neuron position is enough to produce the effect rather that this particular model of noisy adaptation? I also wonder whether other types of frequency adaptation (for instance non-linear "spike" frequency adaptation involving r rather than U) would still lead to this behavior.***
>
> **[Answer 1]** Indeed, our model requires noisy adaptation, but this is biologically very plausible. Firstly, adaptation, as exemplified by SFA in the present study, is widely observed in neural systems, and the biophysics process for generating SFA in neurons is also well documented (see Line 101 - 103) [1]. Second, noises are ubiquitous in neural systems, ranging from the stochastic release of neurotransmitters to the Poisson spiking of neurons. Therefore, it is quite natural to consider noisy adaptation, rather than a fixed SFA strength, when modelling real neural systems.
>
> For the purpose of theoretically analyzing the network dynamics and hence elucidating the generation of Levy flights clearly, we have adopted a simple form of SFA. This simple form of SFA is used in the theoretical neuroscience literature [2]. But, more importantly, the exact form of adaptation is not crucial to generating Levy flights, as long as it has the effect of destabilizing the attractor states of a network to induce long-jump motions. For instance, short-term depression (STD) of neuronal synapses is another way to incur adaptive neuronal activities, and previous studies have shown that a CANN with STD can have the travelling wave state [3]. Thus, noisy STD with appropriate strengths can equally generate Levy flights in attractor networks. We have discussed this in the manuscript (see Line 316 and Line 320- 323).  Similarly, non-linear "spike" frequency adaptation， involving r rather than U， will have the same effect. In fact, by replacing U with r (the right side of Eqn 4 becomes $-V + (m+\sigma_m\xi_m)r$), there is still a theoretical guarantee that Eqn 17 &18 hold true, with only the expressions of $\mu$ and $\gamma$ changed (with $\mu = 1-\tau_vm(1+m)J_0/(4\pi a^2k(1+m)^2+2\rho J^2_0)$ and $\gamma = \sigma_m/(\sqrt{2\pi}am)$) .
>
> The exact form of noises is also not critical, as long as they can induce large fluctuations in the adaptation strength. For instance, by replacing adding noise on adaptation strength directly ($\xi_m$) with adding noise on SFA current (the right side of Eqn 4 becomes $-V + mU + \sigma_V\xi_v$), there is still a theoretical guarantee that Eqn 17 &18 hold true, with only the expressions of $\gamma$ changed ($\gamma = \sqrt{8\pi}ak(1+m)\sigma_V/(mJ_0)$). However, if noises are only added in the dynamics of neural activity, i.e., in Equ.2, they will not generate Levy flights: the network state will only exhibit Brownian motion or travelling wave, depending on the amplitude of the fixed SFA strength. Similarly, if noises are included in the connection pattern between neurons, they will not generate Levy flights, as there is no mechanism to induce intermittent state switches. Typically, for static random connections between neurons, the network will be trapped in a single stationary state, if there is no adaptation to destabilize the attractor.
>
> **[Limitation 2]** ***A second limitation is that the authors make an implicit far-fetched connection between Levy flights in behavior from Drosophila and humans and Levy-flights in neural activity (line 35 to 44). The geographical motion of human is certainly explained by a different phenomenon than the behavior Drosophila, is it therefore ok to suggest that your model is an answer to all this by mentioning it in the intro? I believe it would be more convincing to minimize or clarify this paragraph, at least I do not think that the presented model explains all these behaviors. I do find that the speculation that the model could explain mind wandering is charming, although quite hypothetical. Ideally, this third section should acknowledge that this connection is a speculation or justify it better.***
>
> **[Answer 2]** Thank the reviewer for raising this issue. Indeed, the way of our writing may have misled readers to that we will propose a unified mechanism to explain all Levy flight behaviors, although our motivation of introducing Levy flights across species is to emphasize the importance of studying Levy flights. We will re-write the introduction to clarify this confusion as suggested by the reviewer. Also, as suggested by the reviewer, we will state that our model is a phenomenological one that reproduces the free memory recall experiment in the revised manuscript.
>
> **REFERENCES:**
>
> [1] Gutkin, B., & Zeldenrust, F. (2014). Spike frequency adaptation. Scholarpedia, 9(2), 30643.
>
> [2] P. C. Bressloff. Spatiotemporal Dynamics of Continuum Neural Fields. J. Phys. A, 45, 033001 (2012).
>
> [3] Fung, C. A., Wong, K. M., Wang, H., & Wu, S. (2012). Dynamical synapses enhance neural information processing: gracefulness, accuracy, and mobility. Neural computation, 24(5), 1147-1185.

---

> > ### Comment · Reviewer_FVLu · 2021-08-12
> > **Improved score**
> >
> > The authors appropriately addressed the two limitations that I raised. I assume that they will take that into account when updating the manuscript and I updated my score accordingly.

---

### Official Review · Reviewer_1hsR · 2021-07-15

**Rating:** 7
**Confidence:** 2

**Summary:**

Levy flights are a class of random walks that are efficient search strategies, and human cognitive functions have been shown to exhibit Levy flight characteristics. This work demonstrates that an adaptation of continuous attractor neural networks (CANNs) can generate Levy flight dynamics. Essentially, by setting the parameters of the network near a threshold where the network transitions from Brownian dynamics to traveling wave dynamics, and adding external noise, one can induce the dynamics to randomly switch between Brownian dynamics and traveling wave dynamics, thus mimicking Levy flight dynamics.

**Ethical Concerns:**

I have no ethical concerns.

**Limitations And Societal Impact:**

How would one test whether brains use "spike frequency adaption" to generate Levy flights?

**Main Review:**

This work is the first to study how the brain can generate Levy flight patterns which have been observed experimentally. Some of the interesting theory unpinning this work was previously established in [30,31], where it was shown that depending on the parameters, the CANN will exhibit Brownian dynamics or traveling wave dynamics. The main novelty here is to show that when the parameters are near the threshold between the 2 modes and there is external noise, the network will switch between the 2 modes and they obtain an analytical expression for the Levy exponent. In the conclusion, they propose that short-term depression can destabilize attractor states

Minor comments:

1. Following Eq. (2), should "Fig. 2B" be "Fig. 2A"?
2. Caption of Fig. 2, I believe the authors mean "for clarity" instead of "for clearance".

**Time Spent Reviewing:**

4

---

> ### Author Response · Authors · 2021-08-10
> **Reply to Reviewer 1hsR**
>
> **We acknowledge the encouraging comments of the reviewer about our work. We will correct all typos pointed out by the reviewer in the revised manuscript.**
>
> Here, we would like to restate the significance and novelty of our work, in case they were not clearly described in our manuscript. In this work, we have proposed a neural circuit model to generate Levy flights in the brain. For the first time, we show that an attractor network with noisy adaptation can generate Levy flights. Remarkably, we solved the network dynamics analytically and elucidated the underlying mechanism clearly, that is, noisy adaptation causes the network state to intermittently switch between local and long-jump motions, leading to Levy flights. We then applied the model to reproduce the free memory recall experiment. Considering the biological plausibility and generality of attractor networks and noisy adaptation, we believe that this study is a potentially important contribution to the field, as it lays the foundation for us to further understand the emergence of Lévy flights in more complicated brain cognitive functions and animal foraging behaviors.
>
> **The question raised by the reviewer as “How would one test whether brains use ‘spike frequency adaption’ to generate Levy flights?” is very important. Depending on the experimental data and the progress of experimental techniques, we think we may be able to approach the answer gradually from different levels.**
>
> **Firstly,** we can use our model to reproduce and predict experimental data. Reproducing the free memory recall experiment is an example that has been studied in the current work. Recently, an experimental study just came into our attention [2]. In this work, Pfeiffer and Foster showed that the trajectory of animal position decoded from the sequenced re-activations during sharp wave ripples in hippocampus place cells in awake but immobile rats exhibited Levy flights. Since a CANN is widely used to model the network of place cells, we expect that our model can reproduce this experimental result well and generate predictions on the characteristics of neural activities that are testable in experiments.
>
> **Secondly,** the above studies will only give indirect evidence to our model. To acquire direct evidence, one needs to manipulate SFA at the neuron level and observe its effect on animal behavior. This is much more challenging and relies on the progress of experimental techniques. Biophysically, there are several mechanisms that can cause spike-frequency adaptation [1]. They all include a form of slow negative feedback according to the excitability of the cell. These mechanisms can be summarized as：1) inactivation of depolarizing currents; 2) activity-dependent activation of slow hyperpolarizing or shunting currents, including spike-dependent activation and voltage-dependent activation.  Consider the spike-dependent activation as an example. As pointed out in Line 100-103: ‘SFA could be implemented by the interplay between calcium currents and intracellular calcium dynamics with calcium-gated potassium channels’, we can perform whole cell recording of neurons in a brain slice in vitro to test whether calcium-gated potassium channels could reduce the effect of SFA by applying a channel blocker, e.g., Iberiotoxin (K_{Ca} (BK channel blocker), Apamin (K_{Ca^2} (SK channel blocker), Lei-Dab 7 (high affinity and selective K_{Ca^{2.2}} (SK) blocker). If indeed the SFA effect can be controlled by applying ion channel blockers, we can then observe its effect on animal behavior as predicted by our model. Given the aforementioned experimental study about the Levy-flight pattern in the trajectory of animal position during sharp wave ripples in rats, we could apply a channel blocker to reduce the effect of SFA, and check whether Levy flights are impaired as predicted by our model. This is a challenging experiment, but we hope the rapid progress of experimental techniques will make it feasible soon.
>
> **REFERENCES:**
>
> [1] Benda, J., & Herz, A. V. (2003). A universal model for spike-frequency adaptation. Neural computation, 15(11), 2523-2564.
>
> [2] Pfeiffer, B. E., & Foster, D. J. (2015). Autoassociative dynamics in the generation of sequences of hippocampal place cells. Science, 349(6244), 180-183.

---

### Official Review · Reviewer_JwRM · 2021-07-16

**Rating:** 6
**Confidence:** 1

**Summary:**

The paper discusses the appearance of Levy flights in animal behaviour, and attempts to justify the presence of a "neural mechanism" for this behaviour. The authors begin with a continuous attractor neural network (CANN) model: a linear stochastic partial differential equation involving an "adaptive current" term V, here called "spike frequency adaptation" (SFA). This SFA is the noise adaptation component mentioned in the title, and naturally involves another white noise term multiplied by the activity level U. The authors then approximate the dynamics by condensing the SPDE into a system of SDEs through projection onto two modes. One of the resulting SDEs also contains a multiplicative noise term. This results in a stationary distribution with an asymptotic power law. The appearance of this power law in the motion of these modes is claimed to justify the presence of Levy flights in "neural circuits", and hence in the brain, supported experimentally by a Hebbian-learning attractor network.

**Limitations And Societal Impact:**

The conclusions are not justified up to the standards of mathematical or statistical rigor expected at this venue.

**Main Review:**

I do not believe this is the correct venue for this paper. The problems and motivations appear highly scientific, rooted in applied mathematics and neuroscience, and not related to computer science (or, indeed, to neural networks, despite the name of the model used).

As a mathematician, I recognize that I am not the target audience for this work. From my point of view, when stripped back to its most basic elements, the authors appear to have rediscovered the relationship between multiplicative noise and heavy tails in a particular SPDE model --- see, for example, [1]. Some discussion of this general phenomenon is probably warranted to put the results in context. There is an active area of research surrounding the consequences of this property in ML, including [2,3]. However, I do not see how this paper would be of interest to the ML/statistics community, and think it is better submitted to a neuroscience outlet.

Nevertheless, the paper appears grammatically sound. A few minor issues:
- The CANN SPDE is not ill-posed per se, but Ito/Stratonovich properties of the white noise are not specified
- The intuition and discussion surrounding Levy flights as combining Brownian motion with long jumps is incorrect. Levy flights generally do not exhibit properties analogous to Brownian motion (unless alpha=2). An alpha-stable Levy process is one example of a Levy flight and is purely discontinuous with no Brownian motion part. - "Fokker-Plank" should be written "Fokker-Planck"
- Figure 1 doesn't seem like an especially fair comparison. The variance of the Brownian motion should be adjusted to give a more appropriate match between the Levy flight and the Brownian motion.
- In Figure 4, estimating power laws using curve-fitting to a log-log plot should be avoided, as this approach incurs tremendous bias. An MLE-based approach should be considered instead.


[1] Biró, T. S., & Jakovác, A. (2005). Power-law tails from multiplicative noise. Physical review letters, 94(13), 132302.

[2] Gurbuzbalaban, M., Simsekli, U., & Zhu, L. (2021). The heavy-tail phenomenon in SGD. In International Conference on Machine Learning (pp. 3964-3975). PMLR.

[3] Hodgkinson, L., & Mahoney, M. (2021, July). Multiplicative noise and heavy tails in stochastic optimization. In International Conference on Machine Learning (pp. 4262-4274). PMLR.

**Time Spent Reviewing:**

4

---

> ### Author Response · Authors · 2021-08-10
> **Reply to Reviewer JwRN**
>
> We acknowledge the reviewer for reviewing our paper, but we would like to point out that NeurIPS is the right venue to publish our computational neuroscience paper. There has always been a category named “Neuroscience and Cognitive Science” in NeurIPS to accept papers related to computational neuroscience. Actually, in the early years, NeurIPS (called NIPS back then) was targeted for neuroscience modelling. Nowadays, it still accepts many neuroscience papers each year, e.g., about 140 submissions in 2018 and about 170 submissions in 2019 (https://neuripsconf.medium.com/what-we-learned-from-neurips-2019-data-111ab996462c). We have been publishing and also reviewing neuroscience papers for NeurIPS for many years.
>
> We agree that down to the basic elements, our model does not offer a new mathematical mechanism for generating Levy flights, but this is not the focus of this study. The key contribution of our work is that we elucidate how neural systems generate Levy flights, a phenomenon which has been widely observed in the brain but its underlying neural mechanism remains unclear. Specifically, for the first time, we show that biologically plausible attractor networks with noisy spike frequency adaptation can implement Levy flights, which gives us insight into understanding how optimal information search is achieved in the brain. It is also worth pointing out that our derivation of the Langevin equation with multiplicative noise [1,2] from a realistic neural circuit model (by applying a projection method) is not trivial, as it unveils clearly how Levy flights rely on neuronal properties that can be tested in neurobiology experiments. All these are novel and important contributions to the neuroscience society.
>
> **Replies to other issues raised by the reviewer:**
>
> **[Ans1] on 'The CANN SPDE is not ill-posed per se, but Ito/Stratonovich properties of the white noise are not specified':**
>
> In Supplements (Line 86), we mentioned that Ito integration for white noises is used.
>
> **[Ans2] on 'The intuition and discussion surrounding Levy flights as combining Brownian motion with long jumps is incorrect':**
>
> We agree that the intuitive description that Levy flights are the combination of Brownian motion and long jumps is not rigorous, although it has been conventionally used in the neuroscience and ecology literature. We will modify this in the revised manuscript.
>
> **[Ans3] on 'Figure 1 doesn't seem like an especially fair comparison':**
>
> We thank the reviewer for pointing out the corrections to be made in Fig.1B. For better illustration, we should use two distributions having roughly the same mean value of step sizes. We will modify this in the revised manuscript.
>
> **[Ans4] on 'Estimating power laws using curve-fitting to a log-log plot should be avoided':**
>
> We only used log-log plot in Figure 4, which is for comparison with the experimental data in the same format (comparing to Fig.2C in the reference [3]). In all other Figures (see Fig. 3 A&D&F, Fig. 4 C), we used maximum likelihood estimation (MLE) to test and fit the simulation results with the power-law distribution. We also fit the results with competing distributions to test if power law is the only plausible hypothesis. Results ruled out competing distributions and showed that the fitting of power law distribution is quite good (R-square > 0.95), which supports the power-law distribution of step sizes. We will make the code available on github soon.
>
> **REFERENCES:**
>
> [1] Sakaguchi, H. Fluctuation Dissipation Relation for a Langevin Model with Multiplicative Noise. J. Phys. Soc. Japan 70, 3247–3250 (2001).
>
> [2] Biró, T. S., & Jakovác, A. (2005). Power-law tails from multiplicative noise. Physical review letters, 94(13), 132302.
>
> [3] Rhodes, T. & Turvey, M. T. Human memory retrieval as Lévy foraging. Phys. A Stat. Mech. its Appl. 385, 255–260 (2007).

---

> > ### Comment · Reviewer_JwRM · 2021-08-10
> > **Thank you for your response (and apologies for the error)**
> >
> > Thank you to the authors for your response, and for politely correcting my erroneous assumption of a limited neuroscience audience at NeurIPS. Judging by the other reviews alone, it would appear that my assessment of this paper having limited appeal in this community is in error, and it would be wrong to assert that the paper should be rejected on the grounds of it being off-topic. Clearly I can only review the work in a non-specialist capacity, and to my eye, the paper is written well. I've increased my score to a 6.
> >
> > There are a few minor aspects that I'm not especially fond of, mostly surrounding the discussion surrounding the Lévy flight foraging hypothesis. In particular, the sentence "This study lays the foundation for us to further understand the emergence of Lévy flights in more complicated cognitive functions and ***foraging behaviors***" and its corresponding discussion in Section 4 seems like overreaching to me. My understanding is that the appearance and efficiency of Lévy flights in animal behaviour is very much a separate topic to what is treated here. But this could again come down to my lack of expertise in this particular field.
> >
> > ***"Actually, in the early years, NeurIPS (called NIPS back then) was targeted for neuroscience modelling."***
> > Indeed! For some reason I was under the false impression the conference had since lost its neuroscience audience, but it is good to see this is not the case. Again, my apologies.
> >
> > ***"We agree that down to the basic elements, our model does not offer a new mathematical mechanism for generating Lévy flights, but this is not the focus of this study. The key contribution of our work is that we elucidate how neural systems generate Lévy flights, a phenomenon which has been widely observed in the brain but its underlying neural mechanism remains unclear."***
> >
> > Admittedly, this is not how I interpreted things based on the third paragraph of the conclusions, which seemed to suggest a different mechanism was found from that in the previous literature to date. It appears highly plausible to me that the result is another realization of the usual multiplicative noise/heavy tail connection seen throughout the literature. But I agree the contributions should be focused on the model, rather than a general mathematical mechanism.
> >
> > ***"Specifically, for the first time, we show that biologically plausible attractor networks with noisy spike frequency adaptation can implement Lévy flights, which gives us insight into understanding how optimal information search is achieved in the brain."***
> >
> > Indeed, it is pleasant to see proofs of the appearance of Lévy flights in different communities. I can see the authors have gone to a lot of effort to ensure that this mechanism is well-received within their community and have provided a good deal of intuition.
> >
> > ***"It is also worth pointing out that our derivation of the Langevin equation with multiplicative noise [1,2] from a realistic neural circuit model (by applying a projection method) is not trivial, as it unveils clearly how Lévy flights rely on neuronal properties that can be tested in neurobiology experiments"***
> >
> > I agree that the precise process is not trivial. However it should be clear a priori that the presence of multiplicative noise in (4) will survive each of the projection steps, so the mechanism which provides the heavy tails is the usual one. To be clear, this doesn't diminish any of the contributions here, but the origins of the heavy tails here don't seem as mysterious to me as the paper sometimes seems to suggest.
> >
> > ***"In Supplements (Line 86), we mentioned that Ito integration for white noises is used."***
> >
> > This is true for the stochastic differential equations. However, I was referring to the original SPDE model, where the behaviour of the noise in space is not specified. But since the operations treat the white noise only in its functional form, this is probably unimportant.
> >
> > ***"we used maximum likelihood estimation (MLE) to test and fit the simulation results with the power-law distribution. We also fit the results with competing distributions to test if power law is the only plausible hypothesis"***
> >
> > Thanks for clarifying this. This is very good to see.

---

> > > ### Author Response · Authors · 2021-08-12
> > > **Further Reply to Reviewer JwRN**
> > >
> > > We acknowledge the reviewer for kindly and fairly increasing the score after the discussion. We will also make some statements clearer in the revised manuscript to reduce the misleading, as pointed out by the reviewer.

---

### Official Review · Reviewer_TH9t · 2021-07-20

**Rating:** 7
**Confidence:** 3

**Summary:**

The submission suggests a neural mechanism underlying Levy flights in continuous attractor neural networks (CANN). More specifically, the author(s) proposes that superdiffusion (as observed in free memory retrieval and foraging behavior) occurs in CANN with noisy neural adaptation. External input noise together with adaptation makes the CANN switch between two regimes, local random walks and long-jump motion, separated in the CANN by the traveling wave boundary. Together, these regimes generate superdiffusion. An analytical treatment in reduced dimensionality by a Fokker-Planck approach reveals the two control parameters controlling the (super)diffusion in the model. Implications for free memory retrieval are discussed in a toy model, and as expected, superdiffusion consistent with experimental evidence is observed.

In conclusion, the study proposes an exciting new link to generate in a widely used neural network architecture (CANN) experimentally observed Levy flights using a neuronally plausible mechanism and studies it analytically.

**Limitations And Societal Impact:**

There is no societal impact to discuss.
Limitations:
The authors discuss an extension to STD (short-term depression), but a discussion of the limitations is largely missing.
Here are some potential limitations:
* The derivation in the supplementary material (4) seems to be rather heuristic:
- e.g., the variance of Levy flights is generally diverging; this seems not to be considered here (also not in the supplement).
- the origin of the power-law (scale-invariance) is unclear; I would assume that the traveling waves have a typical scale.
- the switching statistics between traveling wave and Brownian motion are not characterized; it is unclear why the length by which the bump travels should be long-tailed.
- a formal Kramers-Moyal expansion to derive the Fokker-Planck equation is missing.
* If the authors claim their proposed mechanism to generate Levy flights is beyond the mathematical physics literature of anomalous diffusion, it should be made clear what is different from known mechanisms to generate Levy flights (e.g., fractional Fokker-Planck equation). In case the mechanism proposed here can actually be mapped back to a previously described mechanism (which would be totally fine), this should be clearly stated.
* An analysis of the dependence of the Levy-flight phenomenon on the parameters of the Hopfield type model (figure 4) is missing.
* A clear explanation of the link between the model described in figure 3 and the Hopfield-type model (figure 4) is missing. Specifically, while in the CANN, a stationary state was assumed, the Hopfield-type model (figure 4) seems not to be characterized in a stationary state (see above).
I'd be delighted to be convinced that the limitations mentioned above are just resulting from my lack of understanding and are not intrinsic to the work.

**Main Review:**

#postrebuttal:
All major points were addressed.

Originality:
The submission proposes a simple CANN model that is being reduced to two dynamic variables. I am not an expert in the field, but I have not seen that model before; it combines two common 'ingredients' (adaptation and continuous attractor models) in a mathematically tractable model.
(Here seems a related work, but I don't have access to it: https://ses.library.usyd.edu.au/handle/2123/18808
The abstract says: "we develop a new type of two-dimensional neural field model that incorporates refractoriness as nonlinear negative feedback. We construct explicit bump solutions and perform a linear stability analysis, which reveals the emergence of stable bump activity as well as a critical transition from the bump state into propagating waves. Using numerical simulation, we show that the neural field exhibits local propagating patterns with rich dynamics including periodic, rotating and chaotic dynamics. We then show that propagating local patterns undergoing Levy flight emerge from a realistic cortical circuit model and that they can account for a wide range of neural response properties during both spontaneous and stimulus-driven activities."

Even if this work would be similar, the abstract doesn't say anything about an analytical treatment, so I would consider the presented manuscript still to be sufficiently original. Still, the work should be cited, and similarities and differences should be explained.
While previous work on Levy flights and Levy walks (and anomalous diffusion in general) is cited in the discussion, it would help a lot if the mathematical differences would be explained (e.i. why is it not necessary to use a fractional Fokker-Planck equation and fractional calculus here).

Quality:
The model has the appeal of simplicity, so it can be (in a reduced version) be analytically treated using a Fokker-Planck approach, yielding the stationary distributions of s (the separation between bump U and adaptation current V) assuming separation of time scales. Based on that, the phase diagram of 3 dynamics regimes (traveling waves, Brownian motion, and Levy flights) is constructed, and the Levy exponent of the bump movement is determined, which depends on two parameters, 'distance to boundary \mu and noise-to-strength ratio \gamma. Their effect is then compared both analytically and numerically, and numerics and analytical calculations approximately agree.

Major points:
* What is not so clear to me is to what extent the described model actually exhibits a Levy flight. In my (admittedly non-expert) understanding, in Levy flights, increments are drawn from a Levy distribution which asymptotically follows a power law. In the presented manuscript, I don't see any direct evidence that the increments follow a Levy distribution and/or power law. Figure 1B shows an example, but the y-axis is not log-scales, so it is now clear if the decay follows a power law. Moreover, based on the analytical derivation, it is also not fully transparent where such a power law would come from. In the supplement, where the power-law is derived, besides the assumption of a separation of timescales, the assumption is made that "gamma an infinitesimal" (line 121), which, I guess, should read "gamma is infinitesimal". But in the rest of the work, gamma is not infinitesimal -this might be a misunderstanding on my side, clarification would be desirable. I also ran the Matlab code in the supplementary material, but it didn't show how the Levy exponent \alpha reported in the figures was estimated. Also for this, clarification would be desirable.
* Potentially related to the first point, the main model (
* There is no code provided to reproduce the results on free memory retrieval (Figure 4), which seem to be purely numerically in nature.
* An analysis of how the core characteristics of the network (e.g., the Levy exponent \alpha) depend on the model parameters is missing. Do you expect the same behavior as in the model corresponding to figure 3? Based on that, can you predict how the empirically observed \alpha =0.73 would change when you change your model parameters, e.g., input noise \sigma_U, time constants \tau and \tau_V, inhibitory strength k, and Recurrent connection strength J_0? Is it important to have a variance of the SFA strength (\sigma_m) to make the model work? If so, why?
* In the analysis of the Hopfield-like network, it is not clear how \alpha is measured. The manuscript text says: "Further analysis shows that long retrieval intervals are interrupted by bursts of very short intervals and the length of long intervals increases exponentially over time (Fig. 4E). If the length of long intervals increases over time, does it mean the distribution of intervals changes over time, thus also \alpha changes over time? Does it mean there is an aging effect? How is this consistent with the Fokker-Planck analysis, where a stationary distribution of s and z is assumed?


Minor points:
One minor comment on the comparison of analytical and numerical results in Figures 3D and 3F: Please clarify, what the error bars indicate. Are these standard errors of the mean or standard deviation or inter-quartile rate? Also, what ensemble are these error bars calculated over?
* Figure 3D and 3F report numerical results in orange, it is enough to should the orange data points, the lines between the orange points might create the illusion that there is data, but as I understand, there is none. This might also help to make the blue (analytical results) visible.
I'd be delighted to be convinced that the limitations mentioned above are just resulting from my lack of understanding and are not intrinsic to the work.

* It would also be interesting to look at the search efficiency as a function of noise-to-strength \gamma.
* It would demonstrate that as claimed with a larger grid (when the distribution is less truncated), the numerical simulations approach the theoretical predictions.
* Could the spontaneous activity in the Hopfield-like attractor network also be generated from spontaneous recurrent activity instead of external input drive (e.g., chaos as in asymmetric Hopfield-type networks)?
* Figure 3A and Supplementary figure 2C seem to be smoothed, it would be desirable just to display the raw numerical results without any smoothing that might create the illusion of intermediate values where there are none.

Some typos:
* The label in 3E reads "effeciency" but should read "efficiency".
* "Noises in neural adaptation causes" should read "Noise in neural adaptation causes"
* "also termed as anomalous" should read "also termed anomalous"
* "agree with our theoretical analyses very well" should read "agree with our theoretical analysis very well"
* "The neuronal recurrent connections" should read "The recurrent neuronal connections"
* "represent the coordinates of s on the the axes" should read "represent the coordinates of s on the axes"

**Time Spent Reviewing:**

12

---

> ### Author Response · Authors · 2021-08-10
> **Response to Reviewer TH9t**
>
> **Thank you so much for carefully reviewing our paper. We acknowledge your encouraging words about our work. Your comments are very valuable, and we would like to address them in detail in the below.**
>
> **FOR MAJOR POINTS:**
>
> **[Ans1] On the relationship to the work of Qi.**
>
> Thanks for pointing out this important reference we had missed. We checked their studies and would like to clarify the similarity and differences between our work and theirs.
>
> - Similarity: both works tackled the problem of generating Levy flights in neural circuits using computational modeling.
>
> - Differences: the key differences are that we have used a different network model from Qi and hence proposed a different mechanism to generate Levy flights in neural systems. More concretely, we considered an attractor network (in particular, CANNs for theoretical analysis) and elucidated that noisy spike frequency adaptation (SFA) causes the network state to intermittently switch between local motion and long-jump motion, resulting in Levy flights. Notably, both attractor networks and SFA are biologically plausible. On the other hand, Qi et al. considered an E/I balance spiking network with power-law distributed synaptic weights. In their model, the nonlinear negative feedback (refractoriness as they mentioned in the abstract) is not the key to generate Levy flights, but rather the power-law distributed synaptic weights causes Levy flights (see more discussion in **[Ans2]**). Thus, the mechanism of generating Levy flights in our model is fundamentally different from theirs.  Comparing to their model, we believe that ours is more appealing to describe certain brain functions, e.g., the memory (often viewed as attractors) recall process and the Levy flight-like re-activations during sharp wave ripples in hippocampus place cells (often viewed as a CANN at the circuit level) in rats (see [1] for the experiment data).
>
> Furthermore, as pointed out by the reviewer, our model has the advantage of being analytically solvable, which clearly elucidates how Levy flights depend on network properties (model parameters) (see Equ. 16-18).
>
> We will add Qi’s study and related references, and discuss the similarity and differences to our work in the revised manuscript.
>
> **[Ans2] On the fractional FPE:**
>
> Mathematically, there are different ways to generate Levy flights, such as the Langevin equation with Levy stable noise [2] (fractional FPE), and the Langevin equation with multiplicative noise [3] (non-fractional FPE). In Qi’s model, the power-law distributed synaptic weights with Poisson spiking lead to a SDE with Levy noise (corresponding to fractional FPE). While we consider an attractor network with local connections (no need for the power-law assumption on synaptic weights), and SFA in the neural dynamics. By using a projection method to reduce the network model into a two-variable system, we obtain an expression of the Langevin equation with multiplicative Gaussian noise (see Equ. 14 & 15), which does not lead to the fractional FPE.
>
> **[Ans3] On to what extent our model exhibits Levy flight:**
>
> There are several progressive question here proposed by the reviewer. We will answer them one by one:
>
> - As pointed out by the reviewer, Fig.1B does not show directly that the increments follow a power law distribution. We have carried out the log-log plot, which displays a clear linear relationship. We will add this result to Fig.1 in the revised manuscript.
> - Analytical derivation of the power-law distribution of increments can be obtained as follows: as shown in Equ. 14, the increment of $\mathbf{z}$ is determined by $\mathbf{s}$; the dynamics of $\mathbf{s}$ is further given by Equ. 15, whose stationary distribution is solved to satisfy a power law (Equ. 16); this leads to that the increments of $\mathbf{z}$ satisfy the power-law distribution (Equ. 17). To confirm this theoretical analysis, we will add a comparison of the simulation and theoretical results in the revised manuscript.
> - Thank the reviewer for carefully checking the Supp. We made a typo: ‘$\gamma$ an infinitesimal’ at Line 121 in the Supp, which should be ‘$\mathrm{d}t$ is an infinitesimal’.
> - To estimate $\alpha$, we adopted maximum likelihood estimation (MLE) on the histogram of increments using the ‘fitdist’ function in Matlab. Results showed that the fitting is quite good (R-square > 0.95), supporting the power-law distribution of the increments of $\mathbf{z}$. We will make all the code available on github soon.
>
> **[Ans4], On the generalization of theoretical results to Hopfield-like model:**
>
> Indeed, we only carried out simulations to implement the experimental results for free memory recall. We expect, however, that the theoretical analysis on CANNs in Fig. 3 is applicable to the Hopfield-like net that we have considered, when the density of memorized patterns in the Hopfield net is sufficiently large (the Hopfield net can be viewed as a CANN in the continuum limit). This is understandable, as noise adaptation (SFA) can similarly induce intermittent switches between local motions and long-jump motions in the Hopfield-like net. Based on the theoretical analysis on the CANN, we can predict how $\alpha$ varies with model parameters, e.g., from Equ. 18, we see that $\alpha$ increases with $\mu$ (with $\mu=1-m\tau_v/\tau$) and decreases with $\gamma$ (with $\gamma=\sigma_m/\left(2\sqrt{\pi}am\right)$).
>
> **[Ans5], On the importance of the variance of SFA strength:**
>
> Yes, the variance of SFA strength is essential. Without the variance, the network exhibits either Brownian motion or traveling wave, depending on the SFA strength (see discussions in Line 190 - 195). Only when the variance is included, can the network achieve the intermittent switch between two states.
>
> **[Ans6], On the measurement of $\alpha$ in modeling free memory recall:**
>
> We thank the reviewer for pointing out this important issue, which we have not described clearly in the manuscript. In fact, there are two different $\alpha$ calculated in Fig. 4. On one hand, we measured $\alpha$ based on the step sizes of the network movement in the state space (as done in all other parts of the paper), whose value is stable as illustrated in Fig. 4C. On the other hand, to mimic the experiment data (plotting Fig.4 D-F to reproduce Fig. 1a&b and Fig. 2c in the refs [11]), the $\alpha$ value in Fig. 4F is measured based on the temporal interval between two successive retrievals (see descriptions in Line 261 - 269). Moreover, we have imposed the condition that each memory item can only be visited once as required in the human experiment. Under this constraint, the length of long interval indeed increases over time as observed in both the experimental data and our model, which leads to an unstable $\alpha$ value (reported as the slope in Fig. 4 F). In the revised manuscript, we will distinguish these two different $\alpha$ measurements concretely.
>
> **FOR MINOR POINTS:**
>
> **[Ans7] On the clarification of error bars:**
>
> The error bars in Fig. 3 D&F reflect the standard deviations of the estimated $alpha$. Specifically, we run the model with 10 different trials (with initial randomized seeds), and in each trial, we let the network dynamics to evolve for a sufficiently long time. We then calculated the histogram of increments (step sizes) and used maximum likelihood estimation to calculate $\alpha$. The standard deviation of $\alpha$ is obtained by averaging over trials.
>
> **[Ans8] On the numerical plot in Fig. 3 D&F:**
>
> Thanks for the suggestion. The solid lines between the orange points are a little misleading. We do not have data on the lines between the orange points. We will change them to dash lines in the revised manuscript.
>
> **[Ans9] On the search efficiency as a function of $\gamma$:**
>
> We have the result on the search efficiency as a function of $\gamma$ and we will include them in the revised manuscript (probably in the Supp depending on whether we have enough space in the main text).
>
> **[Ans10] On the larger grid for reducing the difference between simulation results and the theoretical results:**
>
> We will carry out simulation using a larger grid to compare the difference between the simulation and theoretical results.
>
> **[Ans11] On the spontaneous activities in the Hopfield-like attractor network:**
>
> Yes, spontaneous activities in the Hopfield-like attractor network can be generated from spontaneous recurrent activities through asymmetric synaptic connections, see, e.g., [4]. In our model, we used the conventional Hebbian learning rule to encode memory items, which lead to symmetric synaptic connections. In such a case, adaptation is needed to destabilize attractors of the network to generate spontaneous travelling activities.
>
> **[Ans12] On the display of Figure 3A and Supplementary Figure 2C:**
>
> Yes, we will change the plot to the raw numerical results without any smoothing in the revised manuscript. The smoothing comes from the matlab “shading interp” function we have used.
>
> **Thank the reviewer for carefully reviewing our paper. We will correct all typos in the revised manuscript. The reviewer also gave many good suggestions on the limitations of our present study, which we will definitely address when revising the manuscript. We hope that our replies can clarified the major concerns of the reviewer.**
>
> **REFERENCES:**
>
> [1] Pfeiffer & Foster, Autoassociative dynamics in the generation of sequences of hippocampal place cells, (2015) Science.
>
> [2] Schertzer, et al, Fractional Fokker–Planck equation for nonlinear stochastic differential equations driven by non-Gaussian Lévy stable noises (2001), Journal of Mathematical Physics.
>
> [3] Sakaguchi. Fluctuation Dissipation Relation for a Langevin Model with Multiplicative Noise, (2001), J. Phys. Soc. Japan.
>
> [4] Zhang. Representation of spatial orientation by the intrinsic dynamics of the head-direction cell ensemble: a theory, (1996), Journal of Neuroscience.

---

> > ### Comment · Reviewer_TH9t · 2021-09-03
> > **all major points addressed, one minor point still left**
> >
> > We thank the authors for the detailed response. All our major comments were addressed. The manuscript reads even better now, congratulations, on that. One minor comment: I agree with reviewer JwRM that it should be clearly pointed out that this manuscript is not proposing a novel mechanism for Levy flights and clearly state previous mathematical literature on the mechanisms described here, just to avoid confusion.

---

> > > ### Author Response · Authors · 2021-09-03
> > > **Response to the additional comment**
> > >
> > > We thank the reviewer again for the feedback. We will definitely make this point clearer in the revised manuscript. And we thank all the reviewers for their effort made during the discussion panel. All your valuable comments indeed help us a lot to improve the work.

---

### Decision · Program_Chairs · 2021-09-27

**Decision:**

Accept (Poster)

**Comment:**

All the reviewers have agreed that the submission is original and interesting with clear contributions to the domain. Authors’ responses also addressed most major concerns. Hence, I recommend an acceptance.

On another note, as reviewer JwRM partially pointed out, using power-laws and Levy flights for modeling stochastic optimization algorithms have attracted considerable attention in other parts of ML as well. In this respect, I suggest the authors to check the papers and discuss in their paper if they think relevant, that reviewer JwRM suggested, as well as the following one, which is directly based on a Levy-driven Langevin equation:

Simsekli, U., Sagun, L., & Gurbuzbalaban, M. (2019, May). A tail-index analysis of stochastic gradient noise in deep neural networks. In International Conference on Machine Learning (pp. 5827-5837). PMLR.

Zhou, P., Feng, J., Ma, C., Xiong, C., & Hoi, S. C. H. (2020). Towards Theoretically Understanding Why Sgd Generalizes Better Than Adam in Deep Learning. Advances in Neural Information Processing Systems, 33.